# A Last Glacial Maximum forcing dataset for ocean modelling

Anne L. Morée[1], Jörg Schwinger[2]

[1]Geophysical Institute, University of Bergen and Bjerknes Centre for Climate Research, Bergen, 5007, Norway

[2]NORCE Climate, Bjerknes Centre for Climate Research, 5007 Bergen, Norway

*Correspondence to*: Anne L. Morée (anne.moree@uib.no)

**Abstract.** Model simulations of the Last Glacial Maximum (LGM, ~21 000 years before present) can aid the interpretation of proxy records, help to gain an improved mechanistic understanding of the LGM climate system, and are valuable for the evaluation of model performance in a different climate state. Ocean-ice only model configurations forced by prescribed atmospheric data (referred to as "forced ocean models") drastically reduce the

computational cost of paleoclimate modelling as compared to fully coupled model frameworks. While feedbacks between the atmosphere and ocean and sea-ice compartments of the Earth system are not present in such model configurations, many scientific questions can be addressed with models of this type. Our dataset supports simulation of the LGM in a forced ocean model set-up, while still taking advantage of the complexity of fully coupled model set-ups. The data presented here are derived from fully coupled paleoclimate simulations of the

Palaeoclimate Modelling Intercomparison Project phase 3 (PMIP3). The data are publicly accessible at the NIRD Research Data Archive at https://doi.org/10.11582/2020.00052 (Morée and Schwinger, 2020). They consist of 2-D anomaly forcing fields suitable for use in ocean models that employ a bulk forcing approach and are optimized for use with CORE forcing fields. The data include specific humidity, downwelling longwave and shortwave radiation, precipitation, wind (v and u components), temperature, and sea surface salinity (SSS). All fields are

provided as climatological mean anomalies between LGM and pre-industrial simulations. These anomaly data can therefore be added to any pre-industrial ocean forcing data set in order to obtain forcing fields representative of LGM conditions as simulated by PMIP3 models. Furthermore, the dataset can be easily updated to reflect results from upcoming and future paleo model intercomparison activities.

## 1 Introduction

The LGM (~21 kya) is of interest to the climate research community because of the relative abundance of proxy data, and because it is the most recent profoundly different climatic state of our planet. For these reasons, the LGM is extensively studied in modelling frameworks (e.g., Menviel et al., 2017; Brady et al., 2012; Otto-Bliesner et al., 2007; Bouttes et al., 2011; Buchanan et al., 2016; Lynch-Stieglitz et al., 2016; Kageyama et al., 2017). Model simulations of the past ocean cannot only provide a method to gain a mechanistic understanding of marine proxy

records, they can also inform us about model performance in a different climatic state of the Earth system (Braconnot et al., 2012). Typical state-of-the-art tools to simulate the (past) Earth system are climate or Earth system models as, for example, used in the Coupled Model Intercomparison Project phase 5 (CMIP5; Taylor et al., 2011). Besides simulating our present climate, these CMIP5 models are also used to simulate past climate states (such as the LGM) in the Palaeoclimate Modelling Intercomparison Project 3 (PMIP3). However, the

computational costs and run-time of such fully coupled model frameworks are a major obstacle for their application to palaeoclimate modelling. Palaeoclimate modelling optimally requires long (thousands to ten thousands of years) simulations in order to provide the necessary time for relevant processes to emerge (e.g. $CaCO_3$ compensation) (Braconnot et al., 2007). Complex fully coupled models can typically not be run into full equilibrium (which requires hundreds to thousands of years of integration) due to computational costs (Eyring et al., 2016). Therefore, the PMIP3 models exhibit model drift (especially in the deep ocean, e.g. Marzocchi and Jansen, 2017). Since significant differences between a (drifting) non-equilibrated state and the equilibrium model state can impede comparison of model results with proxy data, a well equilibrated model with minimal drift is desirable. The 3rd phase of the PMIP project (Braconnot et al., 2012) limits global mean sea-surface temperature drift to under 0.05 K per century and requires the Atlantic Meridional Overturning Circulation to be stable (Kageyama et al., 2018). We refer to a "forced ocean model" as a model of the ocean-sea-ice-atmosphere system in which the atmosphere is represented by prescribed 2-D forcing fields. Such model set-ups have been extensively used in model intercomparison studies such as the Coordinated Ocean-ice Reference Experiments (COREs; Griffies et al., 2009), and more recently in the CMIP6 Ocean Model Intercomparison Project (OMIP; Griffies et al. 2016). A forced ocean model can be used whenever ocean-atmosphere feedbacks are of minor importance and has the advantage of reducing the computational costs – making longer or more model runs feasible. The use of PMIP output in forced ocean modelling is common practice (e.g., Muglia and Schmittner, 2015; Khatiwala et al., 2019). Until now however, there is no standardized dataset available that can be used to easily derive a Last Glacial Maximum model forcing. Therefore, we present 2-D (surface) anomaly fields of CMIP5/PMIP3 experiments 'lgm' (representing the Last Glacial Maximum state of the Earth system) minus 'piControl' (representing the pre-industrial state) calculated from monthly climatological PMIP3 output. The PMIP3 output is the result of global boundary conditions and forcings (such as insolation and ice sheet cover) applied in the fully coupled PMIP3 models (Braconnot et al., 2012). Our dataset (Morée and Schwinger, 2020) is a unique compilation of existing data, processed and reformatted such that it can be readily applied in a forced ocean model framework that uses a bulk forcing approach similar to Large and Yeager (2004). Since this approach has been popularized through coordinated model intercomparison activities (Griffies et al., 2009), a majority of forced ocean models today uses this approach. The 2-D anomaly fields presented here can be added to the pre-industrial forcing of a forced ocean model in order to obtain an atmospheric forcing representative of the LGM. The data are climatological mean anomalies, and as such suitable for equilibrium LGM 'time-slice' modelling of the ocean. In Sect. 2, a general description of the dataset and data sources is provided alongside an overview of the variables (Table 1). The description of the procedure followed to make this dataset (Sect. 3) should support any extension of the dataset with additional (PMIP-derived) variables if needed. The PMIP4 guidelines (Kageyama et al., 2017) can support users in designing a specific model set-up, for example regarding the land-sea mask, trace gas concentrations, river runoff or other conditions and forcing one would want to apply to a model. Limitations of the dataset are discussed in Sect. 4.

**2 General description of the dataset**

The data presented in this article are 2-D anomaly fields of the LGM versus pre-industrial state based on PMIP3 (Braconnot et al., 2012). We note that the PMIP3/CMIP5 pre-industrial state, which is the result of 'piControl'

experiments, represents the year 1850 and is therefore strictly speaking already influenced by anthropogenic forcing (e.g., Eyring et al., 2016). Our anomaly fields can be used as atmospheric LGM forcing fields for ocean-only model set-ups when added to pre-industrial forcing fields (as done by e.g. Muglia and Schmittner, 2015; Khatiwala et al., 2019), and are optimized for use in combination with CORE forcing fields (Griffies et al., 2009).

We note that the CORE forcing is based on modern era (1948-2009) reanalysis and observations, and thus is not a pre-industrial forcing. However, the anthropogenic climate signal contained in these data is relatively small, particularly in comparison to the uncertainties of the LGM-PI anomalies (see below). The basis of the anomaly data is monthly climatological PMIP3 output. Any variables presented on sub-monthly time resolution are therefore time-interpolated. We chose to time-interpolate the variables to their respective time resolution in the

CORE Normal Year Forcing format (CORE-NYF; Large and Yeager, 2004). The anomalies are calculated as the mean of the difference between monthly climatologies of the 'lgm' and 'piControl' PMIP3 model runs. In cases where modelling groups provided more than one ensemble member, we included only the first member in our calculations. The data is the mean anomaly of five PMIP3 models (CNRM-CM5, IPSL-CM5A-LR, GISS-E2-R, MIROC-ESM and MRI-CGCM3: Table 2), as only these models provide output for all variables. A discussion on

the limitations of our dataset is provided in Sect. 4.

The variables are i) near-surface specific humidity, ii) downwelling longwave radiation, iii) downwelling shortwave radiation, iv) precipitation, v) wind (v and u components), vi) near surface temperature, and vii) sea surface salinity (SSS) (Table 1). The SSS anomaly field can be used to adjust SSS restoring in LGM simulations. All variables (Sect. 3.1-7) of the monthly climatological PMIP3 output have been regridded (Table 3, #1), averaged

(Table 3, #2), and differenced (Table 3, #3) to calculate the anomaly fields. Additional procedures for each variable are provided in the respective part of Sect. 3, together with a figure of each variable's annual mean anomaly and model spread. Alongside the lgm-piControl anomaly for each variable, the model spread across all five models is made available. The individual model anomalies for each of the variables are presented in Fig. A1. In order to give the reader the opportunity to compare the anomaly data with typical pre-industrial values for each of the variables,

we provide the multi-model annual mean for the piControl experiment in Fig. A2.

The inter-model disagreement is described for each variable in Sect. 3, and could for example be used to guide adjustments of the amplitude of the forcing anomaly for model tuning purposes. Additionally, proxy-based reconstructions are available for some of the variables, which can constrain potential adjustments to the forcing anomaly fields. We note however that for none of our variables a purely proxy-based global reconstruction exists

– underlining the value of model-based reconstructions. A combination of model and proxy data makes it feasible to create global coverage for air temperatures (e.g., Annan and Hargreaves, 2013), but we are not aware of similar efforts for any of our other variables. Regional proxy-based reconstructions, although mostly quantitative and only over the continents, exist for humidity (e.g., Alexandre et al., 2018), precipitation (e.g., Mendes et al., 2019) as well as wind direction and strength (e.g., Markewich et al., 2015). Regarding ocean proxies, salinity

reconstructions are highly uncertain (Rohling, 2000), but could also provide some constraint to the model data. We leave the decision to the individual modelling groups whether to adjust their forcing fields for their specific application.

All operations were performed with NetCDF toolkits CDO version 1.9.3 (Schulzweida, 2019) or NCO version 4.6.9. The main functions used are documented in Table 3, and referred to in the text at the first occurrence. The

40 atmospheric anomaly data are on a Gaussian grid, with 192×94 (lon×lat) grid-points. The SSS fields are on a

regular 360×180 (lon×lat) grid. Regridding any of the files to a different model grid should be straightforward (e.g., Table 3, #1), as it was ensured that all files contain the information needed for re-gridding. The variables, grid and time resolution are chosen to be compatible with the CORE forcing fields (Large and Yeager, 2004), which have been extensively used in the ocean modelling community as they are the standard in ocean model comparisons (Griffies et al., 2009, 2016). We anticipate that the variables selected here should be useful in different model set-ups as well. We intend to provide a data set that is flexible with respect to the use of different land-ocean masks in different models. Therefore, we account for changes in sea-level (i.e. a larger land area in the LGM), which can affect variables in coastline areas, by applying the following masking procedure: i) masking the multi-model mean anomaly with the maximum lgm land mask across all models, then ii) extrapolating the variable over land using a distance-weighted average (Table 3, #4), and iii) finally masking the data with a present-day land mask (based on the World Ocean Atlas 2013 1° resolution land mask), but with the ocean extended in a 1.5 degrees radius over land. Therefore, our anomaly forcing dataset can likely be used with any pre-industrial land-sea mask. Through following this procedure, the grid points affected by land-sea mask changes are thus filled with the extrapolated model mean anomaly from the LGM coastal ocean. In the case of NorESM-OC (Schwinger et al., 2016), the atmospheric anomaly fields were added to its CORE-NYF fields (Large and Yeager, 2004) to obtain an LGM normal-year forcing, under the assumption of unchanged spatial and temporal variability for the respective variable. Note that the addition of the anomaly fields to the user's own model forcing could lead to physically unrealistic/not-meaningful results for some variables (such as negative precipitation or radiation). This must be corrected for by capping off sub-zero values (Table 3, #5) after addition of the anomaly.

## 3 The variables

### 3.1 Specific humidity anomaly

The mean anomaly of near-surface specific humidity over the five models was time interpolated (Table 3, #6) to a 6-hour time resolution from the monthly climatological PMIP3 output. The annual mean lgm-piControl anomaly field (Fig. 1) shows a global decrease in specific humidity, as expected from decreased air temperatures (Sect. 3.6). The anomaly is most pronounced around the equator, where we see a decrease of $2\text{-}3\times10^{-3}$ kg kg$^{-1}$, while the anomaly is near-zero towards both poles. The model spread of the anomaly shows a disagreement between the PMIP3 models generally in the order of $1\text{-}2\times10^{-3}$ kg kg$^{-1}$, but is larger (up $4\times10^{-3}$ kg kg$^{-1}$) in the northern hemisphere western boundary current regions and close to the Arctic ice edge (Fig. 1).

### 3.2 Downwelling longwave radiation anomaly

The anomaly for surface downwelling longwave radiation is time-interpolated (Table 3, #6) to a daily time resolution. The annual mean anomaly field (Fig. 2) shows globally decreased downwelling longwave radiation in the 'lgm' experiment as compared to the 'piControl' experiment, in the order of 10-30 W m$^{-2}$ over most of the ocean due to a generally cooler atmosphere (Sect. 3.6). The largest anomalies lie close to the northern ice sheets, with up to -90 W m$^{-2}$ lower radiation in the 'lgm' experiment than in the 'piControl' experiment. Ice is likely also the main contributor to the high (60-90 W m$^{-2}$) inter-model spread in North Atlantic and Southern Oceans (Fig.

A3). The remainder of the ocean exhibits a better agreement, with inter-model spreads generally below 20 W m$^{-2}$ (Fig. 2).

### 3.3 Downwelling shortwave radiation anomaly

The surface downwelling shortwave radiation anomaly field is time-interpolated (Table 3, #6) to daily fields as done for downwelling longwave radiation. The annual mean anomaly is especially pronounced around the Laurentide and Scandinavian ice sheets, where strong positive anomalies of over ~30 W m$^{-2}$ exist (Fig. 3). Globally, the annual mean downwelling shortwave radiation anomaly generally falls in a range of -15 to +15 W m$^{-2}$ over the ocean. The anomaly field shows negative anomalies as well positive ones in an alternating spatial pattern approximately symmetrical around the equator in the Pacific basin. The inter-model spread is largest in the North Atlantic region and along the equator (Fig. 3). Due to the large model disagreement of up to 50 W m$^{-2}$ for this variable (Fig. 3), the inter-model spread and mean anomaly are of similar magnitude although a consistent pattern is present in the anomaly field.

### 3.4 Precipitation anomaly and river runoff

The anomaly precipitation presented here is the lgm-piControl anomaly at the air-sea interface and includes both the liquid and solid phases from all types of clouds (both large-scale and convective). The units were converted to mm day$^{-1}$ to comply with the CORE forcing format (causing a deviation from the CF-1.6 convention). The resulting annual mean anomaly generally falls in the range of -2 to 2 mm day$^{-1}$, and is most pronounced along the equator (Fig. 4). The models show a mean increase in precipitation directly south of the equator in the Pacific basin, as well as in the Pacific subtropics off the western North-American coast. The North Atlantic also receives a mean positive precipitation anomaly, offsetting part of the positive salinity anomaly there, which is potentially relevant for the simulation of deep-water formation in this region (Sect. 3.7). Negative mean precipitation anomalies are most pronounced directly north of the equator and north of ~40° N in the Pacific basin as well as in the Atlantic Arctic. The inter-model spread is up to ~5 mm day$^{-1}$ around the equator, likely due to the model disagreement about the sign and location of changes in the inter-tropical convergence zone (Fig. 4). Related to precipitation fluxes, river runoff fluxes also changed between the lgm and piControl model experiments. As land-sea masks and river routing are very model specific we cannot provide a gridded river run-off anomaly. Instead, we provide mean absolute and relative large-scale river runoff changes integrated over ocean basins (North/South Atlantic, North/South Pacific, Indian Ocean, Table 4). These anomalies can be used by modelling groups to scale pre-industrial river runoff. Note that evaporation simulated by a forced ocean model will generally not equal the sum of the prescribed precipitation and river runoff. For long integrations it is therefore necessary to adjust one (or both) of these forcings to close the freshwater balance and avoid salinity drift. We assume that modelling groups employing our anomaly forcing will have such a mechanism suitable for their model in place.

### 3.5 Wind anomalies, u and v components

Both for the u and v component of the wind speed, the lgm-piControl anomaly is time-interpolated to 6-hourly fields. The annual mean meridional wind velocity (v, southerly winds) anomaly shows a pronounced increase (~3-5 m s$^{-1}$) in southerly winds around the NW edge of the Laurentide ice sheet as well as over the NW edge of the

Scandinavian ice sheet (Fig. 5). Alongside that, a pronounced decrease (~ 3-5 m s$^{-1}$) in southerly winds is simulated along the eastern North American coast and the Canadian archipelago. The open ocean anomalies are generally small (at most ±1 m s$^{-1}$). The inter-model spread has no pronounced pattern but is sizable, with ~ 1-5 m s$^{-1}$ disagreement between the PMIP3 models. The mean zonal wind velocity (u, westerly winds) anomaly shows alternating negative and positive anomaly bands with an approximate ±2 m s$^{-1}$ range (Fig. 6). This pattern is stronger in the Northern Hemisphere north of ~45° N. The inter-model spread (~1-3 m s$^{-1}$) has little structure except for the ~4-5 m s$^{-1}$ disagreement in the Southern Ocean south of ~40° S, and the ~3-5 m s$^{-1}$ disagreement in the North Atlantic (Fig. 6). In the Southern Ocean the band of large disagreement in westerly wind speeds reflects the large uncertainty in the simulated position of the southern hemisphere jet stream, both in the pre-industrial and the LGM. This disagreement is reinforced by the fact that shifts in the jet position between pre-industrial and LGM also depend on the simulated expansion of sea-ice (Sime et al. 2016; Fig. A3).

### 3.6 Temperature anomaly

The near-surface atmospheric temperature is time-interpolated to calculate the 6-hourly mean anomaly for temperature. The annual mean anomaly is most pronounced in the North Atlantic, where open ocean anomalies exceed -10 K. Elsewhere, the annual mean temperature anomaly is ~ -2.5 K. There is a clear pattern in the model spread: The models show a large spread (>10 K) north of ~45° N, as well as south of ~50° S (5-10 K), likely due to the disagreement about ice cover (Fig. A3). At lower latitudes, the model spread is generally smaller (0-3 K) (Fig. 7).

### 3.7 Sea surface salinity anomaly

Global mean salinity is initialized in PMIP3 models with a 1 psu higher salinity to account for the concentrating effect of the decrease in sea level (Kageyama et al., 2017). Sea surface salinity however, shows a more variable annual mean lgm-piControl change due to changes in the global hydrological cycle (Fig. 8). The sea surface salinity anomaly is presented on a regular 1x1 grid for ease of use. The resulting annual mean SSS anomaly (Fig. 8) shows an increase in sea surface salinity (~1 psu) over the Southern Ocean south of ~55° S, as well as in the Arctic (>3 psu) and the Northern Indian Ocean (~1 psu). A ~2 psu anomaly is simulated in the Canadian Archipelago, the Labrador Sea and across the North Atlantic between what is now Canada and Europe (Fig. 8). Freshening is simulated close to some continents, and is especially pronounced around Scandinavia (about -3 psu). Simulated ocean circulation can be very sensitive to fresh water forcing and thus SSS, especially in the North Atlantic (e.g. Rahmstorf, 1996; Spence et al., 2008). Application of SSS restoring using the SSS anomaly field should therefore be done with caution and attention to its effects on the meridional overturning circulation. Tuning of the salinity anomaly in important deep-water formation regions of up to about ±1 psu, such as done by for example Winguth et al. (1999), may be required to obtain a satisfactory circulation field in reasonable agreement with proxy data. Such adjustments fall well within the PMIP3 model spread (Fig. 8), and show the current limitations of fully coupled PMIP3 models to simulate the LGM hydrological cycle consistent with proxy records of ocean circulation.

**4 Limitations of the dataset**

The anomaly fields presented here are a model-based 'best-estimate' of the LGM anomaly relative to the pre-industrial state. There are some important limitations to these data related to the temporal resolution, the use of model means, and the fact that we rely on modelling results only.

Proxy data with global coverage are unavailable for most of the variables needed to force stand-alone ocean models. We do not attempt to constrain the anomaly fields using the spatially limited information from available proxy data. Consequently, where PMIP3 models are in disagreement with proxy data, our dataset will be so, too. The limitations (or uncertainty) of the PMIP3 simulations can be seen through the large inter-model spread, which is provided with the anomaly data. This does not preclude the possibility that PMIP3 models collectively (i.e. such
that the model spread is small) disagree with available proxy data. Nevertheless, PMIP3 is the state of the art for modelling of past climates at present (Braconnot et al., 2012; Braconnot and Kageyama, 2015).

By adding multi model mean anomalies to forcing fields, dynamical inconsistencies (e.g. between wind and temperature fields) will be created. This means that the resulting forcing fields do not strictly obey the equations of state/motion. A forcing data set would typically be dynamically consistent if the forcing would be the outcome
of an atmospheric model or an advanced reanalysis. The advantage of using model mean fields is that large anomalies of individual models will be smoothed out where models disagree. We believe that currently a main challenge for paleo modelling activities is to achieve long enough integration times. Therefore, using a single forcing (as opposed to using multiple forcings from individual models) seems to be preferable. Regarding the dynamical inconsistencies, it is important to note that the CORE forcing itself (for which our dataset is optimized)
is a mixture of reanalysis and observational data products and as such not dynamically consistent.

PMIP3 model output is publicly available only as monthly mean fields, which also results in some limitations for the anomaly forcing data set. First, although we interpolate the monthly mean anomaly fields to higher (e.g. 6-hourly) temporal resolution, we implicitly assume that any sub-monthly variability (e.g. the diurnal cycle) is preserved from the preindustrial climate state to the LGM state. We can currently not quantify the implications of
this assumption, but future phases of PMIP (providing simulation output with higher temporal resolution) might alleviate this problem. Second, we are not able to accurately re-reference near-surface temperature and humidity to a different reference height. The CORE bulk forcing method of Large and Yeager (2004) requires near-surface specific humidity and temperature at the same height as the wind forcing (at 10 meters). Humidity and temperature are, however, provided at 2 meters height in PMIP3 (as in most atmospheric data products). A procedure to re-
reference humidity and temperature from 2 to 10 meters (e.g. Large and Yeager 2004) requires input data in higher (sub-daily) time resolution in order to resolve different boundary layer stability regimes. However, for an anomaly forcing, the re-referencing only has an effect if it leads to different temperature/humidity increments under the PI and the LGM state. For the open ocean this is barely the case and taking a climatological anomaly of 2-meter-quantities and apply it at 10 meters height is unproblematic. Over sea ice, however, there could be a larger effect
of the re-referencing (due to a significantly different atmospheric stability in the LGM state), especially regarding the temperature. Our analysis indicates that this is probably the case over the central Arctic Ocean (not shown). For all other regions, we estimate that the error made in applying the re-referencing approach on monthly climatological data resolved data, does not justify its application. In general, the error made by omitting the re-

referencing is much smaller than the uncertainties of the anomalies (i.e. the model spread), particularly at high latitudes.

Regarding the robustness of the dataset, we observe that the inclusion of additional model data only leads to minor changes in the anomalies. An example of this is given by comparing version 1 (Morée and Schwinger, 2019) and the current version 3 (Morée and Schwinger, 2020) of this dataset, as the latter also includes the GISS-E2-R model for the calculation of the anomalies. Indeed, individual model anomalies (Fig. A1) show broad agreement, although the magnitude of the anomaly is less agreed on (as discussed in more detail for the individual variables in Sect. 3).

Despite the limitations described here, we believe that using the mean PMIP3 anomaly of coupled models as forcing is currently the best available option for use in stand-alone ocean models. For this purpose, our dataset provides lgm-piControl anomalies in standardized format for the most common variables used in ocean forcing.

## 5 Data availability

The data are publicly accessible at the NIRD Research Data Archive at https://doi.org/10.11582/2020.00052 (Morée and Schwinger, 2020). The .md5 files contain an md5 checksum, which can be used to check whether changes have been made to the respective NetCDF files.

## 6 Summary and Conclusions

The output of the fully coupled PMIP3 simulations of CNRM-CM5, IPSL-CM5A-LR, MIROC-ESM, MRI-CGCM3 and GISS-E2-R is converted to anomaly datasets intended for use in forced ocean modelling of the LGM. All anomalies are calculated as the difference between the 'lgm' and 'piControl' PMIP3 experiments. In addition, all data are formatted in a way that further conversions (of for example units or the grid) can be applied in a straightforward way. The variables are provided in NetCDF format in separate files, and distributed by the NIRD Research Data Archive (Morée and Schwinger, 2020). A climatological LGM forcing data set can be created for any forced ocean model by addition of the presented 2-D anomaly fields to the model's pre-industrial forcing. This approach enables the scientific community to simulate the LGM ocean state in a forced ocean model set-up. We expect that if additional forcing is needed for a specific model, the same approach as described above can be followed. This process is simplified by providing all main CDO and NCO commands used in creating the dataset (Table 3). All data represent a climatological year, i.e. one annual cycle per variable. The application of the data is thus suitable for 'time-slice' equilibrium simulations of the LGM, and optimised for use with the CORE forcing format (Large and Yeager, 2004).

The uncertainty of our anomaly forcing (approximated by the model spread of the PMIP3 models) is generally of similar magnitude as the multi-model annual mean. The complete attribution of the model spread to specific processes is beyond the scope of this article, but our results show that there is considerable uncertainty involved in the magnitude of the anomaly for all variables presented here. Nevertheless, all mean anomalies show a distinct spatial pattern that we expect to be indicative of the LGM-PI changes. Finally, there is currently no other way to reconstruct most of these variables than model simulations with state-of-the-art Earth system models such as those applied in the PMIP3 experiments. For modelling purposes, the inter-model disagreement of PMIP3 provides the

user with leeway to adjust the amplitude of the forcing (guided by the size of the model spread, which is therefore provided alongside the variables in the dataset). Such adjustments can improve model-proxy data agreements, such as described for salinity in Sect. 3.7.

5 **Appendix A.**

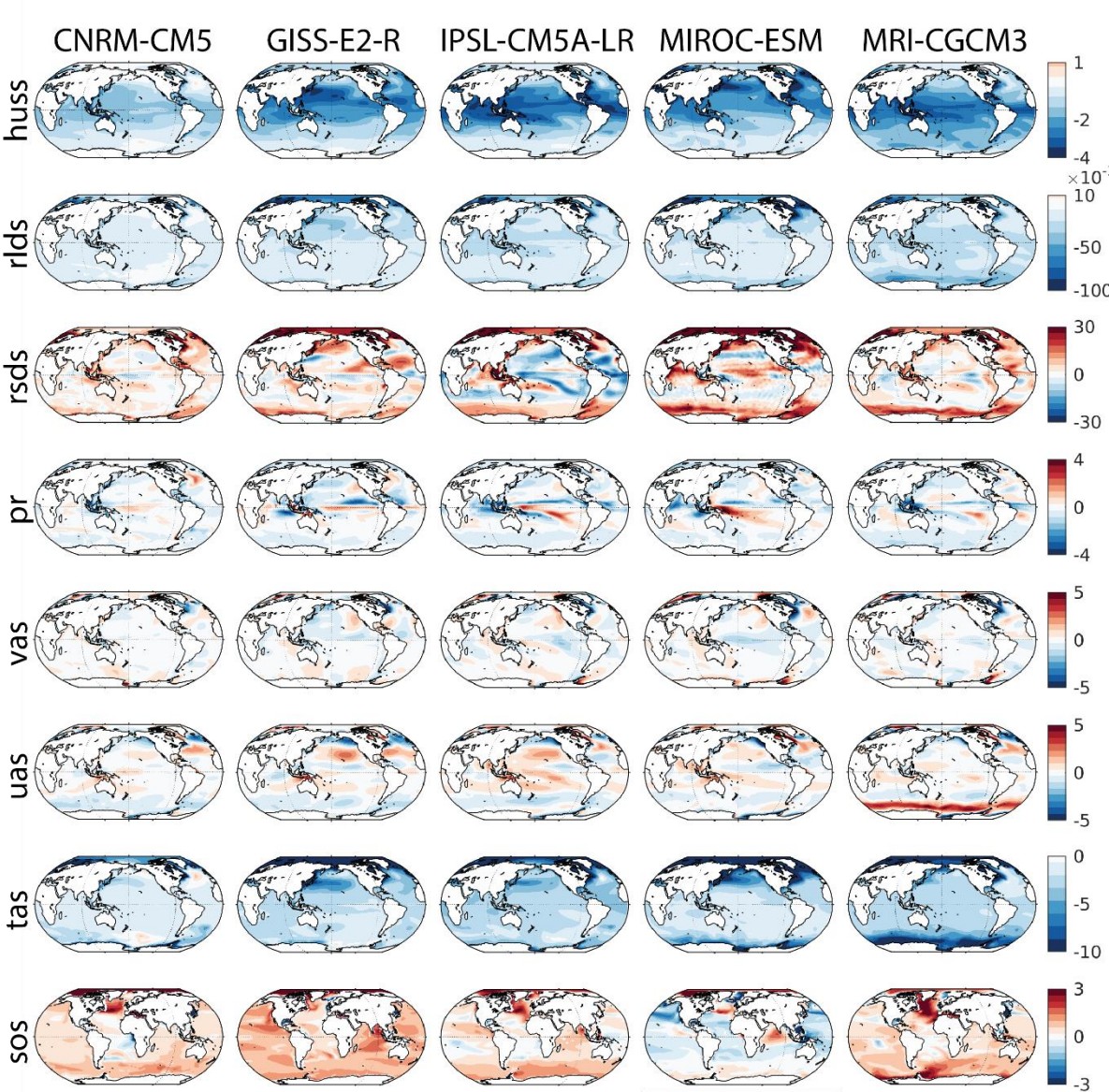

**Figure A1: Annual mean individual model anomalies for each of the variables (see Table 1) and models in the dataset. Units as in the remainder of this manuscript.**

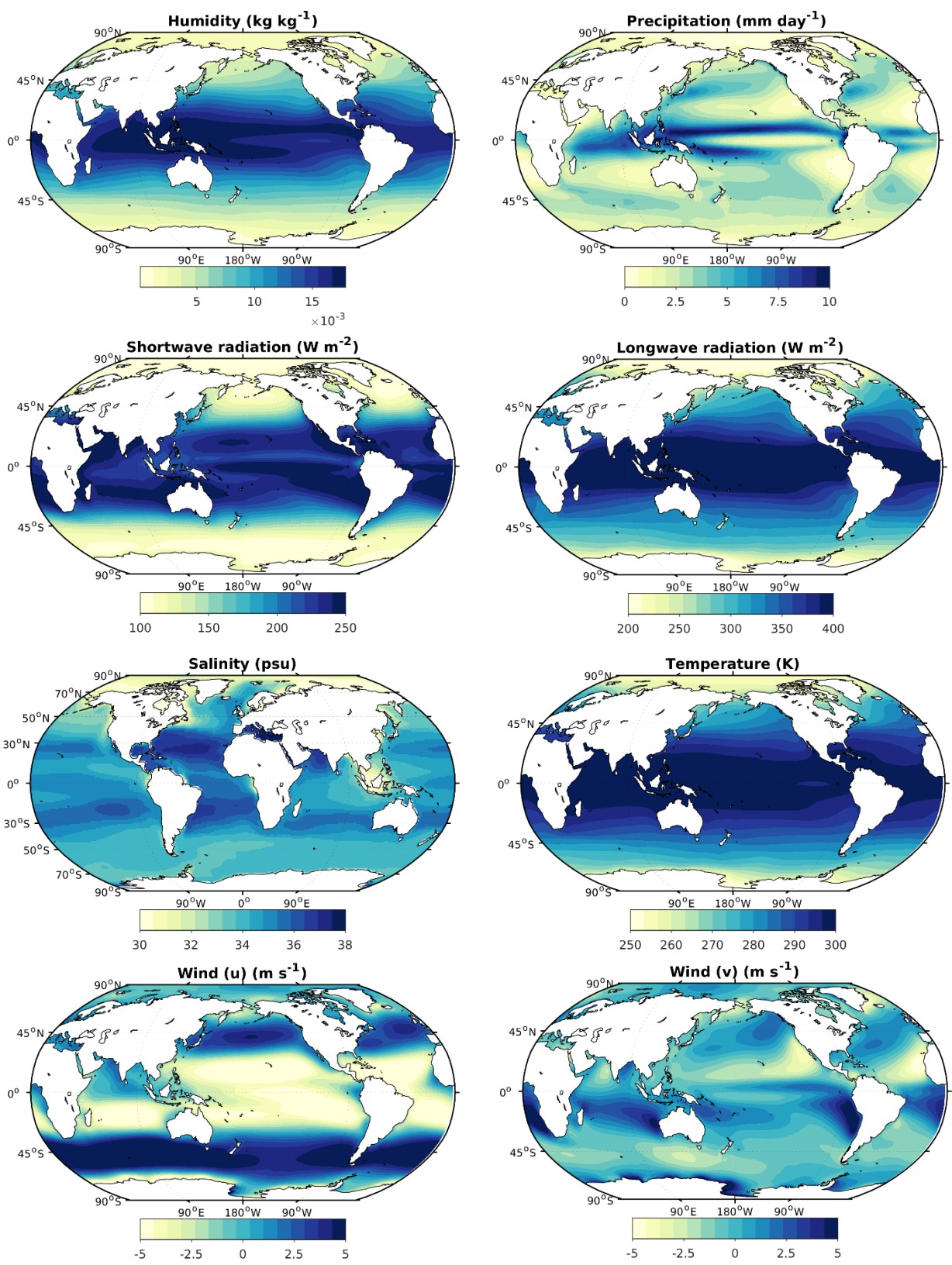

**Figure A2: Annual mean for each of the variables (see Table 1) for the piControl CMIP5/PMIP3 experiment.**

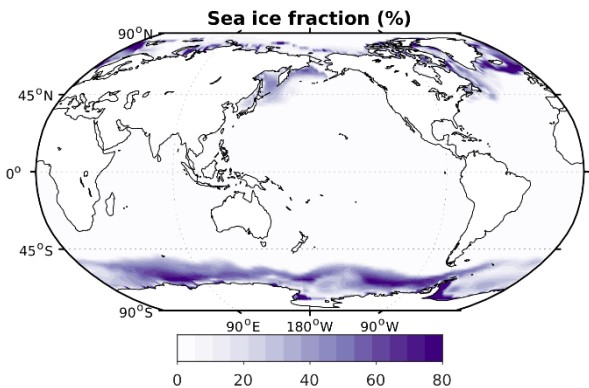

**Figure A3: Annual mean model spread of sea ice fraction.**

**Author contributions.** AM prepared, visualized and analysed the data and wrote the original draft of the manuscript. AM and JS together conceptualized the method and revised the manuscript. JS provided supervision throughout the study.

**Competing interests.** The authors declare that they have no conflict of interest.

**Acknowledgements.** We acknowledge the World Climate Research Programme's Working Group on Coupled Modelling, which is responsible for CMIP, and we thank the climate modelling groups (Table 2) for producing and making available their model output. For CMIP the U.S. Department of Energy's Program for Climate Model Diagnosis and Intercomparison provides coordinating support and led development of software infrastructure in partnership with the Global Organization for Earth System Science Portals. This is a contribution to the Bjerknes

Centre for Climate Research (Bergen, Norway). Storage resources were provided by UNINETT Sigma2 - the National Infrastructure for High Performance Computing and Data Storage in Norway (project number ns2980k). Anne L. Morée is grateful for PhD funding through the Faculty for Mathematics and Natural Sciences of the University of Bergen. Jörg Schwinger acknowledges funding through the Research Council of Norway (project INES, 270061). This study is a contribution to the project "Coordinated Research in Earth Systems and Climate:

Experiments, kNowledge, Dissemination and Outreach" (CRESCENDO; EU Horizon2020 Programme grant no. 641816) which is funded by the European Commission.

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

| Variable description | Units | Resolution (lon×lat), time | Variable name |
|---|---|---|---|
| Specific humidity | kg kg$^{-1}$ | 192×94, 1460 | huss |
| Downwelling longwave radiation | W m$^{-2}$ | 192×94, 365 | rlds |
| Downwelling shortwave radiation | W m$^{-2}$ | 192×94, 365 | rsds |
| Precipitation | mm day$^{-1}$ | 192×94, 12 | pr |
| Wind (u and v components) | m s$^{-1}$ | 192×94, 1460 | uas and vas |
| Temperature | K | 192×94, 1460 | tas |
| Sea surface salinity | psu | 360×180, 12 | sos |

**Table 1: Summary of the data showing variable description, units, format (lon×lat, time), and NetCDF variable name(s). Formats follow CORE conventions (Large and Yeager, 2004). The wind component variables are provided in separate files (Morée and Schwinger, 2020). In each NetCDF file (i.e., for each variable) the model spread is provided alongside the anomaly field named 'variablename_spread'.**

| Model name | Modelling group | Reference | Source data reference |
|---|---|---|---|
| CNRM-CM5 | CNRM-CERFACS (France) | Voldoire et al. (2013) | piControl: Sénési et al. (2014a) <br> lgm: Sénési et al. (2014b) |
| IPSL-CM5A-LR | IPSL (Institut Pierre Simon Laplace, France) | Dufresne et al. (2013) | piControl: Caubel et al. (2016) <br> lgm: Kageyama et al. (2016) |
| MIROC-ESM | MIROC (JAMSTEC and NIES, Japan) | Sueyoshi et al. (2013) | piControl: JAMSTEC et al. (2015a) <br> lgm: JAMSTEC et al. (2015b) |
| MRI-CGCM3 | MRI (Meteorological Research Institute, Japan) | Yukimoto et al. (2012) | piControl: Yukimoto et al. (2015a) <br> lgm: Yukimoto et al. (2015b) |
| GISS-E2-R | NASA/GISS (Goddard Institute for Space Studies, USA) | Schmidt et al. (2014) | piControl: NASA-GISS (2014a) <br> lgm: NASA-GISS (2014b) |

**Table 2: PMIP3 models used in this study**

| # | CDO or NCO command |
|---|---|
| 1 | cdo remapbil,t62grid |
| 2 | cdo ensmean |
| 3 | cdo sub |
| 4 | cdo setmisstodis |
| 5 | ncap2 |
| 6 | cdo inttime |

**Table 3: Package commands applied in this study. Detailed information on these commands can be found in the respective NCO and CDO documentation online. All operations were performed with either CDO version 1.9.3 (Schulzweida, 2019) or NCO version 4.6.9. The complete list of commands is available in the NetCDF files under global attribute 'history'.**

|  | Global | Arctic | N-Atlantic | S-Atlantic | N-Pacific | S-Pacific | Indian |
|---|---|---|---|---|---|---|---|
| PI | 1.20 / | 0.12 / | 0.38 / | 0.14 / | 0.29 / | 0.15 / | 0.12 / |
| $10^9$ kg s$^{-1}$ | 1.41 / | 0.16 / | 0.49 / | 0.17 / | 0.33 / | 0.21 / | 0.16 / |
| (mean/max/min) | 1.07 | 0.07 | 0.32 | 0.13 | 0.25 | 0.09 | 0.08 |
| LGM | 1.23 / | 0.04 / | 0.38 / | 0.15 / | 0.31 / | 0.18 / | 0.17 / |
| $10^9$ kg s$^{-1}$ | 1.41 / | 0.08 / | 0.48 / | 0.17 / | 0.38 / | 0.23 / | 0.25 / |
| (mean/max/min) | 0.93 | 0.004 | 0.20 | 0.12 | 0.24 | 0.14 | 0.12 |
| Relative change | 1.8 / | -70.4 / | -0.7 / | 4.4 / | 6.9 / | 29.6 / | 43.4 / |
| % | 16.3 / | -45.2 / | 20.5 / | 37.7 / | 20.6 / | 59.8 / | 94.5 / |
| (mean/max/min) | -13.2 | -93.9 | -35.7 | -14.2 | -10.2 | 5.6 | 2.6 |

**Table 4: River runoff simulated by the PMIP3 models. Note that only four of the models (CNRM-CM5, IPSL-CM5A-LR, MIROC-ESM, and MRI-CGCM3) provide the necessary output. Arctic is defined as the region north of 65°N, and the eastern boundaries of the Southern Ocean sectors of the Pacific, Atlantic, and Indian are at 70°W, 20°E, and 148°E, respectively.**

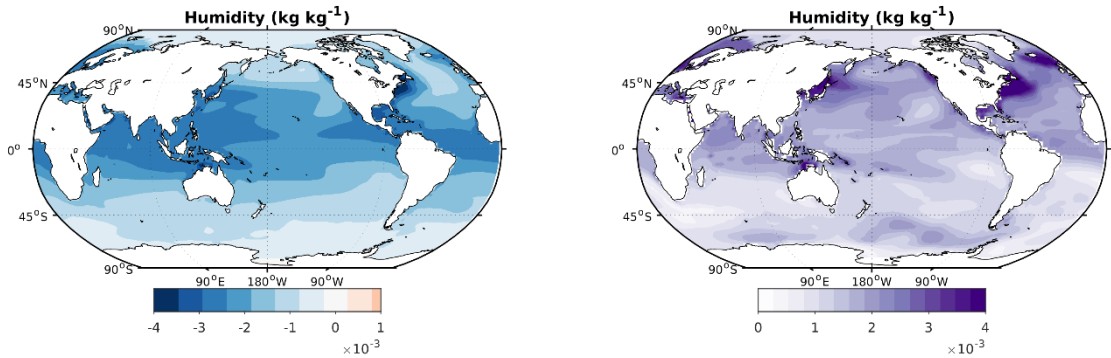

**Figure 1: Annual mean specific humidity lgm-piControl anomaly (left) and model spread (right) in kg kg⁻¹.**

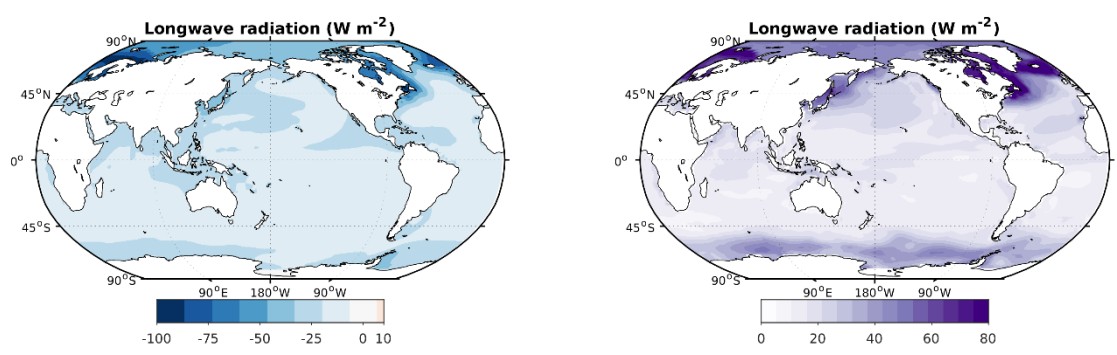

**Figure 2: Annual mean downwelling longwave radiation lgm-piControl anomaly (left) and model spread (right) in W m⁻².**

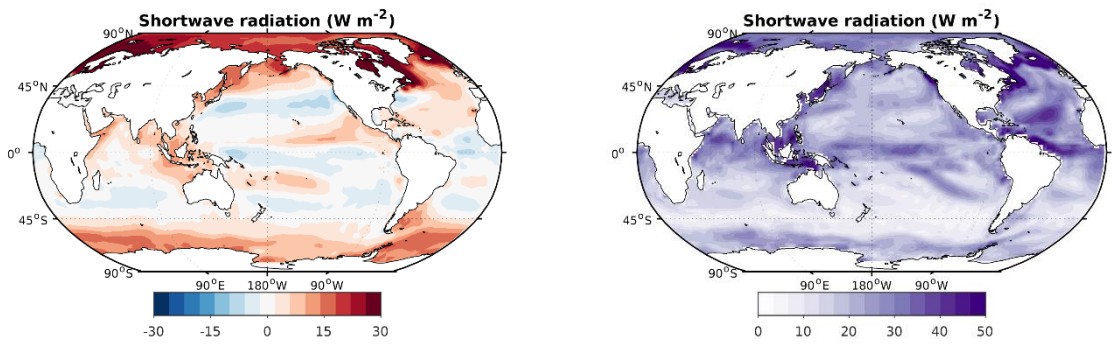

**Figure 3: Annual mean downwelling shortwave radiation lgm-piControl anomaly (left) and model spread (right) in W m⁻².**

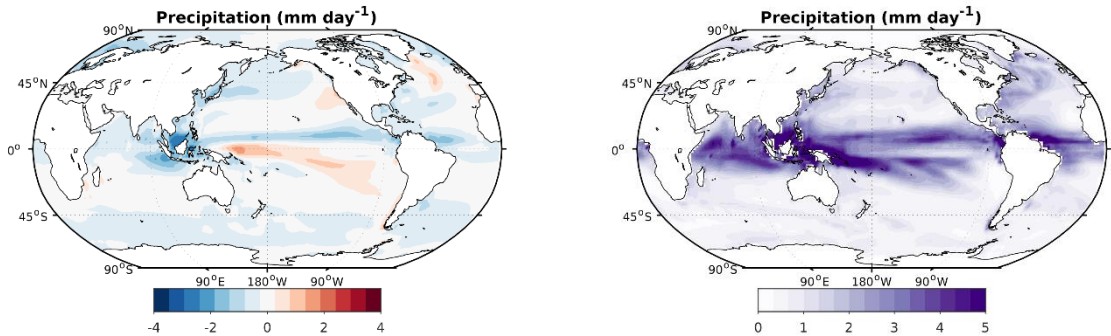

**Figure 4: Annual mean precipitation lgm-piControl anomaly (left) and model spread (right) in mm day⁻¹.**

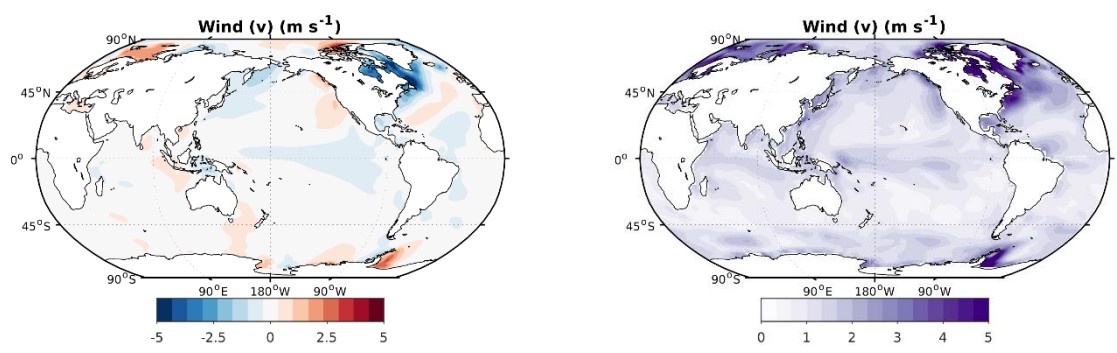

**Figure 5: Annual mean meridional wind velocity lgm-piControl anomaly (left) and model spread (right) in m s⁻¹.**

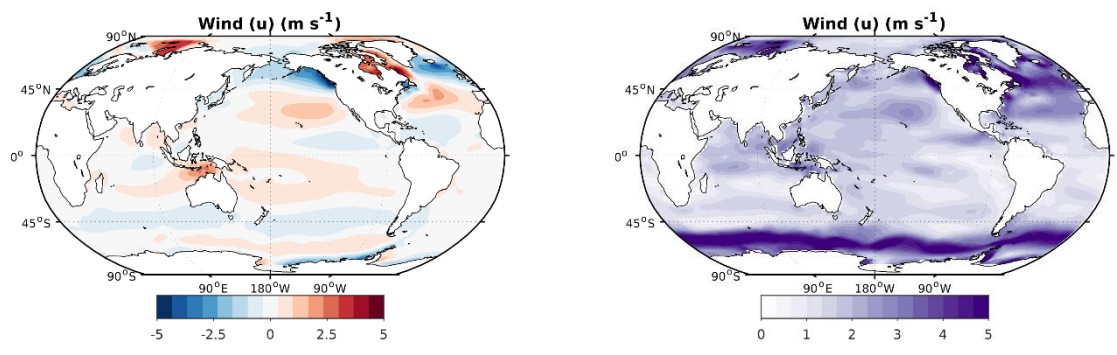

**Figure 6: Annual mean zonal wind velocity lgm-piControl anomaly (left) and model spread (right) in m s⁻¹.**

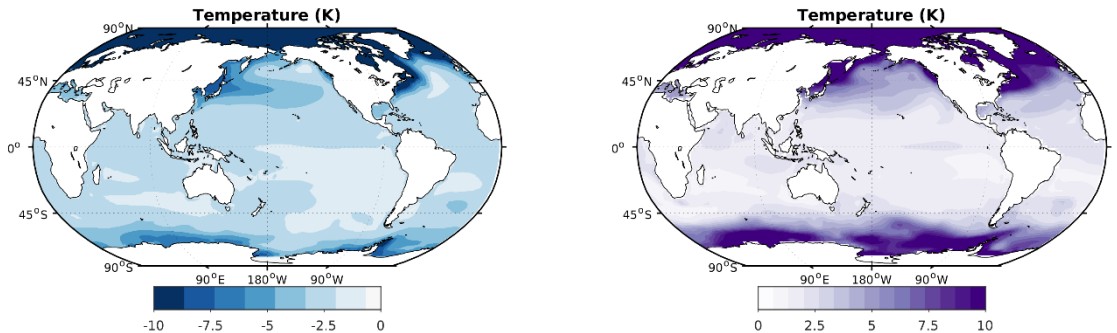

**Figure 7: Annual mean temperature lgm-piControl anomaly (left) and model spread (right) in K.**

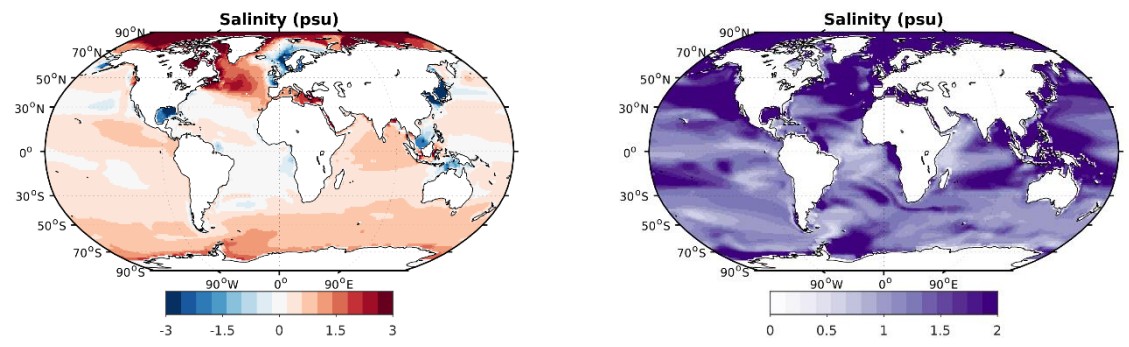

**Figure 8: Annual mean sea surface salinity lgm-piControl anomaly (left) and model spread (right) in psu.**