# Peer review of "A Last Glacial Maximum forcing dataset for ocean modelling"

_Earth System Science Data, 2019_

## Referee Comment (RC1) · Anonymous Referee #1 · 17 Jul 2019

The authors present a model-derived product intended to be used for forcing ocean-only simulations at the Last Glacial Maximum. This product includes a seasonal cycle of atmospheric conditions, which serve as boundary conditions in ocean models, as well as upper-ocean salinity, which is often required as a relaxation boundary condition to avoid drifts.

The manuscript is very clear and well-written, with methods presented in adequate detail. I was easily able to access the product on the linked website. It might be good to tell the user what to do with the *.md5 files. The netcdf files provide useful documentation and appear to have dimensions consistent with their description in the paper.

My main concern with this dataset is that it is not especially novel, as it was derived

very straightforwardly from the PMIP3 archives, which are already publicly available. As such, I am not sure that it will be useful in the future to other studies, given the relative ease with which one can compute anomalies and make the necessary interpolations for running new ocean-only experiments.

I agree that the goal of having a CORE-like set of forcings for the LGM is a useful one, and there are numerous challenges involved in such an exercise. For instance, how does one estimate "normal-year" (intra-annual) forcing typical of a glacial-interglacial anomaly? How should we handle the range of inter-model spreads? How can we include information from available observations? None of these problems is addressed in this work.

More specific concerns:

- Why are some fields interpolated to 6-hourly and others to daily fields?

- While model spread is plotted in the figures, I could not locate this as a variable in the data.

- There may be some grid-scale interpolation artifacts along coastlines (Fig. 1)

- The salinity fields look strange and unlike the figure in the manuscript (Fig. 2) (unless I'm doing something wrong to access the file; I used ncread('Salinity_anomaly_1deg.nc','sos',[1,1,4],[360,180,1]); in MATLAB).

- Please say what NCO is

- There are several thorny issues associated with forcing a model with multi-model means. For one, the fields are no longer dynamically consistent. An implication is that there could be strangely conflicting contributions e.g. to surface salinity from relaxation and precipitation. Second, have these models all been run to equilibrium? Third, computing ensemble means tends to damp uncorrelated variability between members, which reduces the variance of forcing fields. Is there a way to correct for this and generate a "normal year"?

[Figure]

- Are effects from evaporation included in the precipitation file?

- It would be helpful to provide a river runoff file.

- Sec 3.6 line 10: "due to" changes?

—————————————————————

[Figure]

**Humidity anomaly from Moree and Schwinger**

**Fig. 1.** Interpolation artifacts?

**Salinity anomaly from Moree and Schwinger**

**Fig. 2.** Surface salinity

---

## Referee Comment (RC2) · Anonymous Referee #2 · 18 Jul 2019

Review

Title: A Last Glacial Maximum forcing dataset for ocean modelling

Author(s): Anne L. Morée and Jörg Schwinger

MS No.: essd-2019-79

**1. General Comments**

The paper describes the content of the data and its use case in a structured way. The methodology incl. paper references is discussed. The data itself is complete and accessible, checksums are provided.

There are possibilities to improve the paper:

- Improvable Data References:

References for the source data should be added. As part of the terms of use for CMIP5 data, apart from the existing acknowledgement, data collections should be referenced in the article's body and cited in the reference list. For CMIP5 references could be found on the IPCC DDC web page for the AR5 Reference Data Archive: http://www.ipcc-data.org/sim/gcm_monthly/AR5/Reference-Archive.html

- Improvable provenance information:

CMIP5 datasets used should be specified using the full DRS (Data Reference Syntax) including their versions and tracking_ids. All additionally needed datasets like 'psl' and 'ts' (p.3 l. 30) should be specified. An additional table is suggested.

- Question: Why did the authors not use the 3D CMIP5 datasets?

The authors state that only the selected four models provided all the required variables. Could the authors explain why they did not use the 3D fields of the model output but based their study on the interpolated (post-processed) surface variables? The 3D fields were provided by more modeling centers. As the authors apply a vertical interpolation to 10 m height for some variables, the direct model output seems to be better suited as source data. Moreover, the number of models could be increased, on which the data is based. These 3D variables are e.g. 'ta' instead of 'tas', 'hus' ('huss') or 'so' ('sos'). Especially the sea surface salinity anomaly, which is currently based on only two CMIP5 datasets (p.5 ll. 24/25), will become more reliable.

- Reuse of the data:

As the paper on the 'CORE forcing fields' is cited as reference for the usefulness of the chosen spatial-temporal resolution of the provided datasets for common ocean-only model runs, it should be made accessible (e.g. on zenodo) if possible. Alternatively, have the authors used the datasets for the forcing of a second ocean-only model run to show the reusability of the dataset, yet?

- Further reuse of the data:

The authors state that data users could adjust the data using the spread of the CMIP5 model results (p. 6 ll. 29-31). Then the authors need to provide this information in their data.

**2. Specific Comments**

- Please delete 'CMIP-type' as additional characterization of complex fully coupled models, as it is unclear what that means and it does not add information.

- Data files do not contain any history of the applied commands. cdo writes information on the applied commands into the global attribute 'history'. This provides useful information about dataset creation. Why is that not in the file?
- Data files could include more information not only on the above-mentioned history but also on the methodology. I suggest, the authors add the data doi as a reference to the global attributes, which leads the data user to the doi page with further information.
- Why was the unit of precipitation_flux changed from the NetCDF/CF recommended and within CMIP5/PMIP3 used kgm-2s-1 to mm/day? The unit should not be changed if not required.
- The provided datasets same as the CMIP5 datasets should comply with the NetCDF/CF conventions. This seems to be the case, though I did not check it. Then the version of the convention should be specified in the global attributes as described at: http://cfconventions.org/Data/cf-conventions/cf-conventions-1.7/cf-conventions.html.
- The authors should add a sentence on the relation of PMIP3 and CMIP5 (PMIP4 and CMIP6 resp.) for readers less familiar with these large intercomparison projects.
- Is there a reason why the current version 1.9.7 of the cdo package was not used but the old version 1.7.0? Moreover, on the cdo's page 1.7.0 cannot be downloaded (https://code.mpimet.mpg.de/projects/cdo/files). The authors should consider using the current or a more recent version of the cdos.

**3. Technical Corrections**
- CMIP stands for *Coupled* Model Intercomparison Project.
- 'lgm' and 'piControl' are the CMIP5 experiment acronyms. It is confusing and unnecessary to introduce the additional acronyms 'LGM' and 'PI' for them.

---

## Author Comment (AC1) · 14 Aug 2019

**Author Comment to 'Review' in RC1**

1. Explain the use of the *.md5 files

*The *.md5 files contain a md5 checksum which can be used to check whether changes have been made to the respective *.nc files. The md5 checksum of the .nc files should thus be identical to the ones in the *.md5 files. Such checks can be done with several freely available online tools. We will clarify this by explaining it on the dataset webpage of the data repository (the NIRD Research Data Archive).*

2. Comment on the novelty of the dataset

*The reviewer is of course right that the PMIP model results underlying our dataset are publicly available, and that everyone could go to the PMIP archives and repeat our efforts. The assertion that this could be done "very straightforwardly", however, underestimates the amount of work put into the compilation of this dataset considerably. There has been a long series of decisions and considerations involved, starting from compiling data availability across variables and models, to scientific questions of how to treat differences in topography and land-sea mask, and how to re-reference the temperature and humidity fields. The authors expect that the availability of i) a direct visual impression of PI-LGM PMIP3 anomalies and model spread, ii) the clear description of the concept (f.e., CDO/NCO procedure and the idea of anomaly addition to original forcing) and iii) the freely available download of the relevant variables in one standard format will be valuable for future LGM modeling attempts. Indeed, the authors have already received interest from several colleagues about the dataset. Moreover, users inexperienced with PMIP/CMIP/CDO/NCO now have a clear presentation of this opportunity in LGM modeling, which increases the likelihood of modeling attempts by such users. Also, models for which a fully coupled LGM run is not (yet) available, or models that are inherently ocean-only, now have a relatively accessible way to simulate the LGM without needing to prepare such forcing datasets themselves.*

3. How does one estimate NY (intra-annual) forcing typical of G-IG anomaly?

*The construction of a "normal year" forcing (NYF) is indeed not trivial. Large and Yeager (2004; http://opensky.ucar.edu/islandora/object/technotes:434) describe their procedure for the CORE NYF in detail. However, our objective is not to construct a LGM normal year forcing in a strict sense, but to provide LGM-PI anomaly fields that can be used in combination with the existing CORE (and other) forcing fields. If added to the CORE NYF one obtains a LGM "normal year" forcing under the assumption that –beyond changes in the monthly mean state– the temporal and spatial statistics of the forcing fields are the same under LGM conditions. We see that this assumption was not made explicit in our manuscript, and we propose to add text to clarify that we focus on CORE forcing formats, and that the use of an anomaly dataset implies certain assumptions.*

4. How should we handle the range of inter-model spreads?

*The authors address how to use the model spread at the end of the manuscript: '..., all mean anomalies show a distinct spatial pattern that we expect to be indicative of the LGM-PI changes. [...]. For modelling purposes, the inter-model disagreement of PMIP3 provides the user with leeway to adjust the amplitude of the forcing (within the model spread). Such adjustments can improve model-proxy data agreements, such as described for salinity in Sect. 3.7.' Moreover, such adjustments, or 'tuning', of an ocean-only model could inform fully coupled models by revealing model sensitivities. A good example of this is SSS, for which the PI-LGM anomaly is likely too large in the North Atlantic for most PMIP models, pointing to a potential limitation in hydrological cycling of these models (Sect. 3.7).' We acknowledge that we do not know how to objectively handle model spread beyond using it to "justify" model tuning – but think that a discussion of PMIP3 model spread is beyond the scope of the article (p. 6, l. 24-25). The user will be provided with the model spread for each variable in order to improve the usability of the model spreads (see also comment 7).*

5. How can we include information from available observations?

*We believe that for most atmospheric variables (e.g., wind, humidity, radiation), global coverage through LGM proxy data will be never achieved. For model forcing we thus have to rely on model simulations of the variables of interest. Comparison of model results to proxy data that are available (such as estimates of AMOC strength, sea ice extent or productivity for example) is a useful tool for model evaluation (Braconnot et al., 2012). For some atmospheric variables however (e.g., air temperature) one could use proxy data to guide corrections to the model mean anomaly (in addition to the model spread). We leave it to the individual modeling groups to adjust the mean anomalies if considered necessary/needed for their purposes. We will extend the text with information on how the authors think observational/proxy estimates can best be used.*

6. Why are some fields interpolated to 6-hourly and others to daily fields?

*Our intention is to provide the data in CORE forcing format (p. 3, l. 15-16), which could be either NY or interannually varying CORE forcing. For the variables specific humidity, wind components and air temperature CORE forcing is based on the NCEP-reanalysis, which has a standard time resolution of 6 hours (Large and Yeager, 2004). For radiation fluxes, daily is the highest time resolution in the CORE forcing (Large and Yeager, 2004), and therefore used in our dataset. Similarly, for precipitation and SSS, we conform to the CORE forcing standard (monthly time resolution; Large and Yeager, 2004). We will makes this clearer in the manuscript by extending the caption of Table 1, elaborating in lines 15-17 on page 3 and stressing our focus on the CORE format earlier in the manuscript/abstract.*

7. While model spread is plotted in the figures, I could not locate this as a variable in the data.

*We understand this can be a valuable additional variable for the potential user, and will add it to the dataset for all variables named 'variablename_spread'.*

8. There may be some grid-scale interpolation artifacts along coastlines (Fig. 1)

*The most pronounced changes in humidity occur along the coastlines. This is not an artifact of the time or vertical interpolation, but a consequence of the different lgm land-sea mask as compared to the piControl land-sea mask. As some ocean becomes land along most coastlines, a local reduction occurs there in specific humidity.*

*We realize that this is suboptimal, although it is only clearly visible for specific humidity and not relevant for groups that would apply an lgm land-sea mask. However, we want the files to be of use for modeling groups that want to use a pre-industrial land-sea mask as well (e.g. for idealized experiments). To remedy this inconsistency we propose the following approach:*

*1) Mask the multi-model mean anomaly as we present it now by the maximum lgm land area across all models,*

*2) Extrapolate variables over land using a distance-weighted average,*

*3) Mask the data with a decreased present-day land mask (decreased such that we insure that any pre-industrial land-sea mask is covered by the anomaly for all its ocean grid cells).*

*The area affected by land-sea mask changes will thus be filled with the extrapolated model mean anomaly. For humidity, an example of the result is given in the figure below (new anomaly above, old approach below. Note that we added an extra model – see comment 3 in RC2):*

[Figure]

*Coastal effects under the area affected by a changing land-sea mask are thus removed (for all variables). We note that the extrapolatation over the large Arctic continental shelves (which are land during the lgm model runs, and sea during the piControl model runs) leads to artificial structures in the specific humidity anomaly field (top figure). These structures are only relevant if a user applies a present-day land-sea mask, and - we expect - will not cause any problems as the anomaly gradients are similar to those in other regions (f.e. the North Pacific). In addition, no topography correction would be needed for surface temperature, as the field would only be based on open-ocean model data.*

*We tested this approach in NorESM-OC, and see no problems with the initialization caused by the masking.*

9. The salinity fields look strange and unlike the figure in the manuscript (Fig. 2) (unless I'm doing something wrong to access the file; I used ncread('Salinity_anomaly_1deg.nc','sos',[1,1,4],[360,180,1]); in MATLAB).

*The salinity field is the only field provided on a 1x1 degree grid. The field was provided for all grid-cells (extrapolated over land) such that it can be used with any land-ocean mask, as described in the text (Sect. 3.7). As this causes confusion, we propose to follow the same approach as described in our answer to comment 8 – thus providing the data on a decreased pre-industrial land-sea mask.*

10. Please say what NCO is

*NetCDF Operators (NCO) is a NetCDF toolkit (http://nco.sourceforge.net/#Definition) that can be used besides or in addition to CDO. We will clarify this in the manuscript text.*

11. There are several thorny issues associated with forcing a model with multi-model means. For one, the fields are no longer dynamically consistent. An implication is that there could be strangely conflicting contributions e.g. to surface salinity from relaxation and precipitation. Second, have these models all been run to equilibrium? Third, computing ensemble means tends to damp uncorrelated variability between members, which reduces the variance of forcing fields. Is there a way to correct for this and generate a "normal year"?

*i-a) Dynamic consistency: The CORE forcing itself is not dynamically consistent, since it is a blend of different sources (NCEP reanalysis, satellite and surface observations). This is apparently not seen as a major problem in the ocean modelling community (the CORE forcing is widely used, e.g. recently in CMIP6-OMIP). We are not aware either of any study investigating such dynamical inconsistencies in the context of ocean modeling. From this standpoint, we believe that the inconsistencies that are introduced by using a multi-model mean for our anomaly forcing are not a major issue. We will comment on this topic and how it can be prevented for certain variables (see also i-b) in the text.*

*i-b) Salinity relaxation and precipitation: Freshwater balance and salinity relaxation need special attention for multi-centennial forced ocean model simulations. Using the CORE forcing, precipitation is prescribed but evaporation and SSS evolve freely such that imbalances can and will develop. Groups*

*applying the CORE forcing (and our anomaly forcing) will be able to deal with such complications (e.g. enforce freshwater balance globally by adjusting precipitation, balance salinity relaxation to conserve total salinity). In NorESM-OC, this is for example done by enforcing global freshwater-flux balance (by adjusting precipitation with a global correction factor). Also, salinity relaxation is applied such that the global net flux of salt into the surface ocean is zero. Again, these issues occur already when the original CORE forcing is used and are not specific to our anomaly forcing. We will clarify this in the text.*

*ii) Equilibrium: CMIP/PMIP models have been run to equilibrium as much as feasibly possible considering the high costs of computing. The authors comment on this in lines 1-6, p. 2. The output of the different CMIP/PMIP model experiments is considered a reasonable estimate of the past global climate state – which is required for our use. We expect that issues with persistent model drift more likely would arise for interior/deep ocean model fields. Nevertheless, we do acknowledge, that part of the inter-model spread may be explained by differences in the extent to which the respective PMIP model has approached equilibrium. As we comment on equilibration already in l. 1-6 on p. 2, we think no further action is required.*

*iii) Damped uncorrelated variability: We agree with the reviewer that this is an issue with our multi model approach that is not mentioned in our manuscript. The preferred method of using an ensemble of N fully coupled models to force a standalone ocean model would be to create an ensemble of N forcings and run the standalone model for each of the forcings separately. Of course, the drawback of this approach is the N-fold increase in computational resources (cpu and storage). We believe that for the sake of achieving long integrations of the LGM ocean state, it is justifiable to use a multi model mean anomaly. In addition, the anomalies are generally small (< 10 %) as compared to the forcing field. We see that it would be good to mention these considerations explicitly in our manuscript, and we will do so in a new version.*

*In general, we acknowledge that it is important to mention the sources of error coming from the use of multi-model means and will extend the text with comments on the main issues.*

12. Are effects of evaporation included in the precipitation file?

*The variable 'pr' presented in the dataset represents the total amount of precipitation (liquid and solid phases, and from all types of clouds – both large-scale and convective)). Evaporation is thus not included in the precipitation fluxes, and should be calculated by the model itself based on evaporative forcing (through f.e. temperature and humidity). We will improve clarity on this in the section on precipitation.*

13. It would be helpful to provide a river runoff file.

*The authors found that the CMIP river runoff variable (friver, 'Water Flux into Sea Water From Rivers (kg m-2 s-1)') is only available for CNRM-CM5, IPSL-CM5A-LR, MIROC-ESM and MRI-CGCM3 for the lgm. Besides that, the differences in land-sea masks of these models are problematic as river outlets may end up off-coast or on land when applied in a forced ocean model. Averaging over such 'point-sources' will create a little meaningful product in our opinion, as the river outlets vary*

*considerably between models. In NorESM-OC, we route the preindustrial river runoff to the nearest ocean grid-cell, but such solutions are very model dependent, and we expect that modelling groups devise a suitable solution for their specific model for the treatment of runoff. We will comment on river runoff in the precipitation section in order to address the relevance of river routing in LGM setups.*

14. Sect. 3.6 line 10 'due to' changes?

*'due to' is indeed missing here – we will add it to the sentence. Thank you for noting this mistake.*

---

## Author Comment (AC2) · 14 Aug 2019

**Author Comment to 'Review' in RC2**

**General Comments**

1. Improvable Data References:

References for the source data should be added. As part of the terms of use for CMIP5 data, apart from the existing acknowledgement, data collections should be referenced in the article's body and cited in the reference list. For CMIP5 references could be found on the IPCC DDC web page for the AR5 Reference Data Archive: http://www.ipcc-data.org/sim/gcm_monthly/AR5/Reference-Archive.html

*Thank you for making us aware of more specific references that we could use to acknowledge the modeling groups and their specific experiments. We will extend Table 2 to include the experiment and model specific references.*

2. Improvable provenance information:

CMIP5 datasets used should be specified using the full DRS (Data Reference Syntax) including their versions and tracking_ids. All additionally needed datasets like 'psl' and 'ts' (p.3 l. 30) should be specified. An additional table is suggested.

*We will add the the original 'version_history' and 'tracking_id' global attributes of each of the models for both the lgm and piControl runs to the respective variable in our dataset. The CMIP variables 'psl' and 'ts' were used for the re-referencing to 10m height. We will specify their names in the manuscript, such that it is clear which CMIP variables were used for the re-referencing.*

3. Question: Why did the authors not use the 3D CMIP5 datasets?

The authors state that only the selected four models provided all the required variables. Could the authors explain why they did not use the 3D fields of the model output but based their study on the interpolated (post-processed) surface variables? The 3D fields were provided by more modeling centers. As the authors apply a vertical interpolation to 10 m height for some variables, the direct model output seems to be better suited as source data. Moreover, the number of models could be increased, on which the data is based. These 3D variables are e.g. 'ta' instead of 'tas', 'hus' ('huss') or 'so' ('sos'). Especially the sea surface salinity anomaly, which is currently based on only two CMIP5 datasets (p.5 ll. 24/25), will become more reliable.

*The atmospheric 3D fields of CMIP/PMIP are provided on pressure levels. The geopotential height corresponding to the atmospheric pressure level closest to the earth's surface (the 1000 hPa pressure level) is generally ~150-200m above the ocean (CMIP variable 'zg'). The use of the 3D fields on the level of the air-sea interface would thus mean an extrapolation over many tens of meters height for all of the atmospheric variables in our dataset. We expect that the native 2 and 10 meter fields calculated by each model (internally consistent) will provide a better approximation of the near-surface state. The re-referencing of temperature and specific humidity to 10 m done by us is a minor correction compared*

*to this. The choice of surface fields indeed forces us to reduce the number of models we can base our data on. We noticed however that we can add the GISS-E2-R model (which was left out before because we selected r1i1p1 ensembles only) to our dataset. We will add the GISS-E2-R output to our dataset, which will make all atmospheric variables be based on five models instead of the previous four.*

*For sea surface salinity, the use of the ocean 3D field would indeed be an improvement. It would make it possible to use four of the five atmospheric models (namely CNRM-CM5, GISS-E2-R, MIROC-ESM and MRI-CGCM3). We decided in addition to calculate the monthly climatological sea surface salinity based based on the 'mon' 3D salinity (so) output for IPSL-CM5A-LR, as we have not found the 'monClim' IPSL-CM5A-LR data for piControl variable 'so'. In this way, all variables in a new version of the dataset are based on the same 5 models.*

4. Reuse of the data:

As the paper on the 'CORE forcing fields' is cited as reference for the usefulness of the chosen spatial-temporal resolution of the provided datasets for common ocean-only model runs, it should be made accessible (e.g. on Zenodo) if possible. Alternatively, have the authors used the datasets for the forcing of a second ocean-only model run to show the reusability of the dataset, yet?

*The Large and Yeager (2004) report is publicly available at the NCAR/UCAR "OpenSky"-repository: (http://opensky.ucar.edu/islandora/object/technotes:434). We will add this URL to the reference. Experiments with different versions of the CORE forcing data have been done by many modelling groups before – see for example Griffies et al. (2009). The CORE forcing is also used for the Ocean Model Intercomparison Project (OMIP) within CMIP6 (Griffies et al., 2015).*

5. Further reuse of the data:

The authors state that data users could adjust the data using the spread of the CMIP5 model results (p. 6 ll. 29-31). Then the authors need to provide this information in their data.

*We understand this can be a valuable additional variable for the potential user, and will add it to the dataset for all variables.*

**Specific comments**
6. Please delete 'CMIP-type' as additional characterization of complex fully coupled models, as it is unclear what that means and it does not add information.

*We will remove this wording.*

7. Data files do not contain any history of the applied commands. cdo writes information on the applied commands into the global attribute 'history'. This provides useful information about dataset creation. Why is that not in the file?

*The authors wanted to provide clean files to the user alongside the detailed procedure description in the manuscript. As the reviewer points out, the exact procedure is indeed saved in the history (by both CDO and NCO) and could be useful for the potential user. We will keep the full history off all files as well as their appended file history as a global attribute in an updated version of the dataset.*

8. Data files could include more information not only on the above-mentioned history but also on the methodology. I suggest, the authors add the data doi as a reference to the global attributes, which leads the data user to the doi page with further information.

*We will add a global attribute 'references' with the ESSDD manuscript DOI and the NIRD Research Data Archive dataset DOI. As the NIRD Research Data Archive does not allow reservation of DOI's, we can not know before publishing the dataset online what the DOI of the dataset will be. However, we are able to refer to the previous version and state that the user should check for newer versions of the dataset. Similarly, we can not know the final DOI of an ESSD article, so we can only refer to the ESSDD article.*

9. Why was the unit of precipitation_flux changed from the NetCDF/CF recommended and within CMIP5/PMIP3 used kg m-2 s-1 to mm/day? The unit should not be changed if not required.

*The 1979-2000 GXGXS Precipitation Climatology employed for the CORE forcing is in mm/d (Large and Yeager, 2004), and the authors wish to present a dataset that can be easily used in combination with a models' original CORE forcing. We will clarify this in Sect. 3.4 and will provide the conversion factor (86400) used such that the units can be set back to kg m-2 s-1 without data loss. The conversion makes the units of our variable pr deviate from the CF-1.6 convention, which we will note in the manuscript.*

10. The provided datasets same as the CMIP5 datasets should comply with the NetCDF/CF conventions. This seems to be the case, though I did not check it. Then the version of the convention should be specified in the global attributes as described at: http://cfconventions.org/Data/cf-conventions/cf-conventions-1.7/cf-conventions.html.

*We will use one of the available online CF compliance checkers to make sure that the dataset follows NetCDF/CF conventions, and will add the version of the convention as a global attribute in the updated dataset.*

11. The authors should add a sentence on the relation of PMIP3 and CMIP5 (PMIP4 and CMIP6 resp.) for readers less familiar with these large intercomparison projects.

*We will clarify the terms PMIP and CMIP in the introduction of the manuscript.*

12. Is there a reason why the current version 1.9.7 of the cdo package was not used but the old version 1.7.0? Moreover, on the cdo's page 1.7.0 cannot be downloaded (https://code.mpimet.mpg.de/projects/cdo/files). The authors should consider using the current or a more recent version of the cdos.

*The somewhat older 1.7.0 version of CDO gives to our knowledge no different results than later versions for the functions we applied to make our dataset. We however see that the use of the most up-to-date version of CDO is desirable, and we are able to use CDO version 1.9.3 on our systems to remake the dataset. We did a test, and see no differences in the result when using 1.9.3 as compared to version 1.7.0.*

Technical Corrections

13. CMIP stands for *Coupled* Model Intercomparison Project.

*Thank you for noting this mistake – we will correct it in the manuscript.*

14. 'lgm' and 'piControl' are the CMIP5 experiment acronyms. It is confusing and unnecessary to introduce the additional acronyms 'LGM' and 'PI' for them.

*We chose to use LGM and PI for readability, but as this causes confusion we will remove these acronyms from the manuscript.*

*References*

*Griffies, S. M., A. Biastoch, C. Böning, F. Bryan, G. Danabasoglu, E. P. Chassignet, M. H. England, R. Gerdes, H. Haak, R. W. Hallberg, W. Hazeleger, J. Jungclaus, W. G. Large, G. Madec, A. Pirani, B. L. Samuels, M. Scheinert, A. Sen Gupta, C. A. Severijns, H. L. Simmons, A. M. Treguier, M. Winton, S. Yeager and J. Yin, 2009: Coordinated Ocean-ice Reference Experiments (COREs). Ocean Modelling, 26, 1–46.*

*Griffies, S. M., Danabasoglu, G., Durack, P. J., Adcroft, A. J., Balaji, V., Böning, C. W., Chassignet, E. P., Curchitser, E., Deshayes, J., Drange, H., Fox-Kemper, B., Gleckler, P. J., Gregory, J. M., Haak, H., Hallberg, R. W., Heimbach, P., Hewitt, H. T., Holland, D. M., Ilyina, T., Jungclaus, J. H., Komuro, Y., Krasting, J. P., Large, W. G., Marsland, S. J., Masina, S., McDougall, T. J., Nurser, A. J. G., Orr, J. C., Pirani, A., Qiao, F., Stouffer, R. J., Taylor, K. E., Treguier, A. M., Tsujino, H., Uotila, P., Valdivieso, M., Wang, Q., Winton, M., and Yeager, S. G.: OMIP contribution to CMIP6: experimental and diagnostic protocol for the physical component of the Ocean Model Intercomparison Project, Geosci. Model Dev., 9, 3231-3296, https://doi.org/10.5194/gmd-9-3231-2016, 2016.*

---

## Author Response (AR1)

Dear Editor, Dear reviewers,

Thank you for dedicating your time to review our work. We have replied in our Author Response below to 'Review' in RC1 and 'Review' in RC2. This document ends with the marked-up manuscript, showing the changes we have made to the original version. Besides these files, a new version of the dataset is now available at https://doi.org/10.11582/2019.00019.

Yours sincerely,

Anne Morée and Jörg Schwinger

**Author Comment to 'Review' in RC1**

1. Explain the use of the *.md5 files

*The *.md5 files contain an md5 checksum, which can be used to check whether changes have been made to the respective *.nc files. The md5 checksum of the .nc files should thus be identical to the ones in the *.md5 files. Such checks can be done with several freely available online tools.*

**Changes made in the manuscript: An explanation is provided in Sect. 4 on Data Availability.**

2. Comment on the novelty of the dataset

*The reviewer is of course right that the PMIP model results underlying our dataset are publicly available, and that everyone could go to the PMIP archives and repeat our efforts. The assertion that this could be done "very straightforwardly", however, underestimates the amount of work put into the compilation of this dataset considerably. There has been a long series of decisions and considerations involved, starting from compiling data availability across variables and models, to scientific questions of how to treat differences in topography and land-sea mask, and how to re-reference the temperature and humidity fields. The authors expect that the availability of i) a direct visual impression of PI-LGM PMIP3 anomalies and model spread, ii) the clear description of the concept (f.e., CDO/NCO procedure and the idea of anomaly addition to original forcing) and iii) the freely available download of the relevant variables in one standard format will be valuable for future LGM modeling attempts. Indeed, the authors have already received interest from several colleagues about the dataset. Moreover, users inexperienced with PMIP/CMIP/CDO/NCO now have a clear presentation of this opportunity in LGM modeling, which increases the likelihood of modeling attempts by such users. Also, models for which a fully coupled LGM run is not (yet) available, or models that are inherently ocean-only, now have a relatively accessible way to simulate the LGM without needing to prepare such forcing datasets themselves.*

**Changes made in the manuscript: None**

3. How does one estimate NY (intra-annual) forcing typical of G-IG anomaly?

*The construction of a "normal year" forcing (NYF) is indeed not trivial. Large and Yeager (2004; http://opensky.ucar.edu/islandora/object/technotes:434) describe their procedure for the CORE NYF in detail. However, our objective is not to construct a LGM normal year forcing in a strict sense, but to provide LGM-PI anomaly fields that can be used in combination with the existing CORE (and other) forcing fields. If added to the CORE NYF one obtains a LGM "normal year" forcing under the assumption that –beyond changes in the monthly mean state– the temporal and spatial statistics of the forcing fields are the same under LGM conditions. We see that this assumption was not made explicit in our manuscript, and we propose to add text to clarify that we focus on CORE forcing formats, and that the use of an anomaly dataset implies certain assumptions.*

**Changes made in the manuscript: We added additional text in the abstract and Sect. 2 to clarify our intensions (optimized presentation for use with CORE forcing) and assumptions (unchanged spatial and time variability).**

4. How should we handle the range of inter-model spreads?

*The authors address how to use the model spread at the end of the manuscript: '..., all mean anomalies show a distinct spatial pattern that we expect to be indicative of the LGM-PI changes. [...]. For modelling purposes, the inter-model disagreement of PMIP3 provides the user with leeway to adjust the amplitude of the forcing (within the model spread). Such adjustments can improve model-proxy data agreements, such as described for salinity in Sect. 3.7.' Moreover, such adjustments, or 'tuning', of an ocean-only model could inform fully coupled models by revealing model sensitivities. A good example of this is SSS, for which the PI-LGM anomaly is likely too large in the North Atlantic for most PMIP models, pointing to a potential limitation in hydrological cycling of these models (Sect. 3.7).' We acknowledge that we do not know how to objectively handle model spread beyond using it to "justify" model tuning – but think that a discussion of PMIP3 model spread is beyond the scope of the article (p. 6, l. 24-25). The user is provided with the model spread for each variable in a new version of the dataset, in order to improve the usability of the model spreads (see also comment 7).*

**Changes made in the manuscript: Sect. 2 now includes two sentences describing the availability of the model spread alongside the forcing anomaly fields, and what we expect it can be used for. The caption of Table 1 has also been extended to clarify this. We made a new version of the dataset (version 2) that includes the model spread (https://doi.org/10.11582/2019.00019).**

5. How can we include information from available observations?

*We believe that for most atmospheric variables (e.g., wind, humidity, radiation), global coverage through LGM proxy data will be never achieved. For model forcing we thus have to rely on model simulations of the variables of interest. Comparison of model results to proxy data that are available (such as estimates of AMOC strength, sea ice extent or productivity for example) is a useful tool for model evaluation (Braconnot et al., 2012). For some atmospheric variables however (e.g., air temperature) one could use proxy data to guide corrections to the model mean anomaly (in addition to the model spread). We leave it to the individual modeling groups to adjust the mean anomalies if considered necessary/needed for their purposes. We extended the text with information on how the authors think observational/proxy estimates can best be used.*

**Changes made in the manuscript: We extended Sect. 2 with a comment on proxy data as a potential anomaly constraint together with the model spreads.**

6. Why are some fields interpolated to 6-hourly and others to daily fields?

*Our intention is to provide the data in CORE forcing format (p. 3, l. 15-16), which could be either NY or interannually varying CORE forcing. For the variables specific humidity, wind components and air temperature CORE forcing is based on the NCEP-reanalysis, which has a standard time resolution of 6 hours (Large and Yeager, 2004). For radiation fluxes, daily is the highest time resolution in the CORE forcing (Large and Yeager, 2004), and therefore used in our dataset. Similarly, for precipitation and SSS, we conform to the CORE forcing standard (monthly time resolution; Large and Yeager, 2004). We made this clearer in the manuscript by extending the caption of Table 1, and stressing our focus on the CORE format earlier in the manuscript/abstract.*

**Changes made in the manuscript: The caption of Table 1 and the text in Sect. 2 and the abstract are extended to stress our intension to provide CORE format data (see also our reply to comment 3).**

7. While model spread is plotted in the figures, I could not locate this as a variable in the data.

*We understand this can be a valuable additional variable for the potential user, and added it to the dataset for all variables named 'variablename_spread'.*

**Changes made in the manuscript: No changes were made in the manuscript, but a new version of the dataset (version 2) was made (https://doi.org/10.11582/2019.00019) that included the model spread for each variable (with name variable_spread).**

8. There may be some grid-scale interpolation artifacts along coastlines (Fig. 1)

*The most pronounced changes in humidity occur along the coastlines. This is not an artifact of the time or vertical interpolation, but a consequence of the different lgm land-sea mask as compared to the piControl land-sea mask. As some ocean becomes land along most coastlines, a local reduction occurs there in specific humidity.*
*We realize that this is suboptimal, although it is only clearly visible for specific humidity and not relevant for groups that would apply an lgm land-sea mask. However, we want the files to be of use for modeling groups that want to use a pre-industrial land-sea mask as well (e.g. for idealized experiments). To remedy this inconsistency we propose the following approach:*
*1) Mask the multi-model mean anomaly as we present it now by the maximum lgm land area across all models,*
*2) Extrapolate variables over land using a distance-weighted average,*
*3) Mask the data with a decreased present-day land mask (decreased such that we insure that any pre-industrial land-sea mask is covered by the anomaly for all its ocean grid cells).*
*The area affected by land-sea mask changes is thus filled with the extrapolated model mean anomaly. Coastal effects under the area affected by a changing land-sea mask are thus removed for all variables. We note that the extrapolatation over the large Arctic continental shelves (which are land during the lgm model runs, and sea during the piControl model runs) leads to artificial structures in the specific humidity anomaly field (top figure). These structures are only relevant if a user applies a present-day land-sea mask, and - we expect - will not cause any problems as the anomaly gradients are similar to those in other regions (f.e. the North Pacific). In addition, no topography correction is now needed for surface temperature, as the field is now only based on open-ocean model data. We tested this approach in NorESM-OC, and see no problems with the initialization caused by the masking.*

**Changes made in the manuscript: Sect. 2 now includes an explanation of the masking. In the new version of our dataset (version 2), we provide all atmospheric fields with a decreased pre-industrial land mask where the anomaly is set to zero. We include the maximum lgm and decreased 'pre-industrial' masks in each data file such that the user can readily see which part of the data are extrapolated values.**

9. The salinity fields look strange and unlike the figure in the manuscript (Fig. 2) (unless I'm doing something wrong to access the file; I used ncread('Salinity_anomaly_1deg.nc','sos',[1,1,4],[360,180,1]); in MATLAB).

*The salinity field is the only field provided on a 1x1 degree grid. The field was provided for all grid-cells (extrapolated over land) such that it can be used with any land-ocean mask, as described in the text*

*(Sect. 3.7). As this causes confusion, we now provide the data in masked form (see our answer to comment 8).*

**Changes made in the manuscript: We removed the extrapolation procedure from Sect. 3.7 as this is now explained for all variables in Sect. 2. In the new version of the dataset (version 2), no smoothing is needed anymore for sea surface salinity (as we now use more models, see our reply to the other reviewer), so this is also removed from the text.**

10. Please say what NCO is

*NetCDF Operators (NCO) is a NetCDF toolkit (http://nco.sourceforge.net/#Definition) that can be used besides or in addition to CDO.*

**Changes made in the manuscript: At the introduction of CDO and NCO in Sect. 2, we now clarify that each of these are toolkits to handle NetCDF data.**

11. There are several thorny issues associated with forcing a model with multi-model means. For one, the fields are no longer dynamically consistent. An implication is that there could be strangely conflicting contributions e.g. to surface salinity from relaxation and precipitation. Second, have these models all been run to equilibrium? Third, computing ensemble means tends to damp uncorrelated variability between members, which reduces the variance of forcing fields. Is there a way to correct for this and generate a "normal year"?

*i-a) Dynamic consistency: The CORE forcing itself is not dynamically consistent, since it is a blend of different sources (NCEP reanalysis, satellite and surface observations). This is apparently not seen as a major problem in the ocean modelling community (the CORE forcing is widely used, e.g. recently in CMIP6-OMIP). We are not aware either of any study investigating such dynamical inconsistencies in the context of ocean modeling. From this standpoint, we believe that the inconsistencies that are introduced by using a multi-model mean for our anomaly forcing are not a major issue.*

*i-b) Salinity relaxation and precipitation: Freshwater balance and salinity relaxation need special attention for multi-centennial forced ocean model simulations. Using the CORE forcing, precipitation is prescribed but evaporation and SSS evolve freely such that imbalances can and will develop. Groups applying the CORE forcing (and our anomaly forcing) will be able to deal with such complications (e.g. enforce freshwater balance globally by adjusting precipitation, balance salinity relaxation to conserve total salinity). In NorESM-OC, this is for example done by enforcing global freshwater-flux balance (by adjusting precipitation with a global correction factor). Also, salinity relaxation is applied such that the global net flux of salt into the surface ocean is zero. Again, these issues occur already when the original CORE forcing is used and are not specific to our anomaly forcing.*

*ii) Equilibrium: CMIP/PMIP models have been run to equilibrium as much as feasibly possible considering the high costs of computing. The authors comment on this in lines 1-6, p. 2. The output of the different CMIP/PMIP model experiments is considered a reasonable estimate of the past global climate state – which is required for our use. We expect that issues with persistent model drift more likely would arise for interior/deep ocean model fields. Nevertheless, we do acknowledge, that part of the inter-model spread may be explained by differences in the extent to which the respective PMIP model has approached equilibrium. As we comment on equilibration already in l. 1-6 on p. 2, we think no further action is required.*

*iii) Damped uncorrelated variability: We agree with the reviewer that this is an issue with our multi model approach that is not mentioned in our manuscript. The preferred method of using an ensemble of N fully coupled models to force a standalone ocean model would be to create an ensemble of N forcings and run the standalone model for each of the forcings separately. Of course, the drawback of this approach is the N-fold increase in computational resources (cpu and storage). We believe that for the sake of achieving long integrations of the LGM ocean state, it is justifiable to use a multi model mean anomaly. In addition, the anomalies are generally small (< 10 %) as compared to the forcing field. We see that it would be good to mention these considerations explicitly in our manuscript.*

*In general, we acknowledge that it is important to mention the sources of error coming from the use of multi-model means.*

**Changes made in the manuscript: Sect. 2 has been extended to include a comment on dynamical inconsistencies and dampening of uncorrelated variability.**

12. Are effects of evaporation included in the precipitation file?

*The variable 'pr' presented in the dataset represents the total amount of precipitation (liquid and solid phases, and from all types of clouds – both large-scale and convective)). Evaporation is thus not included in the precipitation fluxes, and should be calculated by the model itself based on evaporative forcing (through f.e. temperature and humidity).*

**Changes made in the manuscript: We made it explicit in Sect. 3.4 that the precipitation anomaly excludes evaporation.**

13. It would be helpful to provide a river runoff file.

*The authors found that the CMIP river runoff variable (friver, 'Water Flux into Sea Water From Rivers (kg m-2 s-1)') is only available for       CNRM-CM5, IPSL-CM5A-LR, MIROC-ESM and MRI-CGCM3 for the lgm. Besides that, the differences in land-sea masks of these models are problematic as river outlets may end up off-coast or on land when applied in a forced ocean model. Averaging over such 'point-sources' will create a little meaningful product in our opinion, as the river outlets vary considerably between models. In NorESM-OC, we route the preindustrial river runoff to the nearest ocean grid-cell, but such solutions are very model dependent, and we expect that modelling groups devise a suitable solution for their specific model for the treatment of runoff.*

**Changes made in the manuscript: We extended Sect. 3.4 with a comment on river runoff and our recommendations to follow PMIP guidelines.**

14. Sect. 3.6 line 10 'due to' changes?

*'due to' is indeed missing here. Thank you for noting this mistake.*

**Changes made in the manuscript: None - as the topography-corrected temperature anomaly is not provided in version 2 of our dataset due to the masking and extrapolation procedure (see our reply to comment 8), this sentence is not present in the manuscript anymore.**

**Author Comment to 'Review' in RC2**

**General Comments**

1. Improvable Data References:

References for the source data should be added. As part of the terms of use for CMIP5 data, apart from the existing acknowledgement, data collections should be referenced in the article's body and cited in the reference list. For CMIP5 references could be found on the IPCC DDC web page for the AR5 Reference Data Archive: http://www.ipcc-data.org/sim/gcm_monthly/AR5/Reference-Archive.html

*Thank you for making us aware of more specific references that we could use to acknowledge the modeling groups and their specific experiments.*

**Changes made in the manuscript: We extended Table 2 to include the experiment specific references and updated the reference list accordingly.**

2. Improvable provenance information:

CMIP5 datasets used should be specified using the full DRS (Data Reference Syntax) including their versions and tracking_ids. All additionally needed datasets like 'psl' and 'ts' (p.3 l. 30) should be specified. An additional table is suggested.

*It would indeed be good to complement the dataset with the original 'version_history' and 'tracking_id' global attributes of each of the models for both the lgm and piControl experiments. The CMIP variables 'psl' and 'ts' were used for the re-referencing to 10m height.*

**Changes made in the manuscript: In the new version of the dataset (version 2), all variables now have the tracking_id numbers for each of the piControl and lgm output files, as well as the version_history in their global attributes. We specified the names 'psl' and 'ts' in Sect. 3.1, such that it is clear which CMIP variables were used for the re-referencing from 2 to 10 meters.**

3. Question: Why did the authors not use the 3D CMIP5 datasets?

The authors state that only the selected four models provided all the required variables. Could the authors explain why they did not use the 3D fields of the model output but based their study on the interpolated (post-processed) surface variables? The 3D fields were provided by more modeling centers. As the authors apply a vertical interpolation to 10 m height for some variables, the direct model output seems to be better suited as source data. Moreover, the number of models could be increased, on which the data is based. These 3D variables are e.g. 'ta' instead of 'tas', 'hus' ('huss') or 'so' ('sos'). Especially the sea surface salinity anomaly, which is currently based on only two CMIP5 datasets (p.5 ll. 24/25), will be more reliable.

*The atmospheric 3D fields of CMIP/PMIP are provided on pressure levels. The geopotential height corresponding to the atmospheric pressure level closest to the earth's surface (the 1000 hPa pressure level) is generally ~150-200m above the ocean (CMIP variable 'zg'). The use of the 3D fields on the level of the air-sea interface would thus mean an extrapolation over many tens of meters height for all of the atmospheric variables in our dataset. We expect that the native 2 and 10 meter fields calculated by*

*each model (internally consistent) will provide a better approximation of the near-surface state. The re-referencing of temperature and specific humidity to 10 m done by us is a minor correction compared to this. The choice of surface fields indeed forces us to reduce the number of models we can base our data on. We noticed however that we can add the GISS-E2-R model (which was left out before because we selected r1i1p1 ensembles only) to our dataset.*

*For sea surface salinity, the use of the ocean 3D field would indeed be an improvement. It would make it possible to use four of the five atmospheric models (namely CNRM-CM5, GISS-E2-R, MIROC-ESM and MRI-CGCM3). We decided in addition to calculate the monthly climatological sea surface salinity based based on the 'mon' 3D salinity (so) output for IPSL-CM5A-LR, as we have not found the 'monClim' IPSL-CM5A-LR data for piControl variable 'so'.*

**Changes made in the manuscript: We clarify that the new dataset is based on five models for all variables (Sect. 2). The dataset has been updated (https://doi.org/10.11582/2019.00019), i.e. version 2) to include the GISS-E2-R model for all atmospheric variables. Sea surface salinity is now calculated based on the 3D variable 'so' from PMIP3 (and we updated Table 1 accordingly). The whole dataset is therefore based on the same five models in version 2 of the dataset.**

4. Reuse of the data:

As the paper on the 'CORE forcing fields' is cited as reference for the usefulness of the chosen spatial-temporal resolution of the provided datasets for common ocean-only model runs, it should be made accessible (e.g. on Zenodo) if possible. Alternatively, have the authors used the datasets for the forcing of a second ocean-only model run to show the reusability of the dataset, yet?

*The Large and Yeager (2004) report is publicly available at the NCAR/UCAR "OpenSky"-repository: (http://opensky.ucar.edu/islandora/object/technotes:434). Experiments with different versions of the CORE forcing data have been done by many modelling groups before – see for example Griffies et al. (2009). The CORE forcing is also used for the Ocean Model Intercomparison Project (OMIP) within CMIP6 (Griffies et al., 2015).*

**Changes made in the manuscript: We added the DOI url to the report to the Large and Yeager (2004) reference.**

5. Further reuse of the data:

The authors state that data users could adjust the data using the spread of the CMIP5 model results (p. 6 ll. 29-31). Then the authors need to provide this information in their data.

*We understand this can be a valuable additional variable for the potential user.*

**Changes made in the manuscript: See also our reply to comment 8 of the other reviewer: We elaborate on the model spread in Sect. 2. The new version of the dataset (i.e., version 2) contains the model spread for each variable alongside the anomaly.**

**Specific comments**
6. Please delete 'CMIP-type' as additional characterization of complex fully coupled models, as it is unclear what that means and it does not add information.

*Thank you for noting this.*

**Changes made in the manuscript: We removed this wording in the Introduction.**

7. Data files do not contain any history of the applied commands. cdo writes information on the applied commands into the global attribute 'history'. This provides useful information about dataset creation. Why is that not in the file?

*The authors wanted to provide clean files to the user alongside the detailed procedure description in the manuscript. As the reviewer points out, the exact procedure is indeed saved in the history (by both CDO and NCO) and could be useful for the potential user. We will keep the full history off all files as well as their appended file history as a global attribute in an updated version of the dataset.*

**Changes made in the manuscript: None. However, the new version of the dataset (i.e., version 2) now contains a full history of the CDO and NCO commands used.**

8. Data files could include more information not only on the above-mentioned history but also on the methodology. I suggest, the authors add the data doi as a reference to the global attributes, which leads the data user to the doi page with further information.

*As the NIRD Research Data Archive does not allow reservation of DOI's, we cannot know before publishing the dataset online what the DOI of the dataset will be. However, we are able to refer to the previous version and state that the user should check for newer versions of the dataset. Similarly, we cannot know the final DOI of an ESSD article, so we can only refer to the ESSDD article.*

**Changes made in the manuscript: We added a global attribute 'references' with the ESSDD manuscript DOI and the NIRD Research Data Archive dataset DOI (dataset version 1, as we cannot know the new DOI before publication in the archive) and a note to check for updates.**

9. Why was the unit of precipitation_flux changed from the NetCDF/CF recommended and within CMIP5/PMIP3 used kg m-2 s-1 to mm/day? The unit should not be changed if not required.

*The 1979-2000 GXGXS Precipitation Climatology employed for the CORE forcing is in mm/d (Large and Yeager, 2004), and the authors wish to present a dataset that can be easily used in combination with a models' original CORE forcing.*

**Changes made in the manuscript: The deviation from the CF-1.6 convention is noted in Sect. 3.4. The conversion factor is provided in the global attribute 'Conventions' in the new version of the dataset.**

10. The provided datasets same as the CMIP5 datasets should comply with the NetCDF/CF conventions. This seems to be the case, though I did not check it. Then the version of the convention should be specified in the global attributes as described at: http://cfconventions.org/Data/cf-conventions/cf-conventions-1.7/cf-conventions.html.

*We used one of the available online CF compliance checkers to make sure that the dataset follows NetCDF/CF conventions. Note that the units of precipitation do not follow the convention, in order to follow the CORE format convention (see our reply to comment 9).*

**Changes made in the manuscript: We added the version of the convention (CF-1.6) as a global attribute in the new version of the dataset.**

11. The authors should add a sentence on the relation of PMIP3 and CMIP5 (PMIP4 and CMIP6 resp.) for readers less familiar with these large intercomparison projects.

*We see the need to further clarify the PMIP3-CMIP5 connection in the introduction of the manuscript.*

**Changes made in the manuscript: We added a sentence in the Introduction explaining what PMIP3 is as part of CMIP5.**

12. Is there a reason why the current version 1.9.7 of the cdo package was not used but the old version 1.7.0? Moreover, on the cdo's page 1.7.0 cannot be downloaded (https://code.mpimet.mpg.de/projects/cdo/files). The authors should consider using the current or a more recent version of the cdos.

*The somewhat older 1.7.0 version of CDO gives to our knowledge no different results than later versions for the functions we applied to make our dataset. We however see that the use of the most up-to-date version of CDO is desirable, and we are able to use CDO version 1.9.3 on our systems to remake the dataset. We did a test, and see no differences in the result when using 1.9.3 as compared to version 1.7.0.*

**Changes made in the manuscript: CDO version 1.9.3 and NCO version 4.6.9 were used to make version 2 of the dataset, and noted as a global attributes in the dataset. The text (caption Table 3 and Sect. 2) is updated accordingly.**

**Technical Corrections**

13. CMIP stands for *Coupled* Model Intercomparison Project.

*Thank you for noting this mistake.*

**Changes made in the manuscript: We corrected it in the Introduction of the manuscript.**

14. 'lgm' and 'piControl' are the CMIP5 experiment acronyms. It is confusing and unnecessary to introduce the additional acronyms 'LGM' and 'PI' for them.

*We chose to use LGM and PI for readability, but decided to change this as it causes confusion.*

**Changes made in the manuscript: We removed these PI and LGM acronyms from the manuscript wherever they were directly referring to the 'lgm' and 'piControl' PMIP/CMIP experiments.**

*References*

*Griffies, S. M., A. Biastoch, C. Böning, F. Bryan, G. Danabasoglu, E. P. Chassignet, M. H. England, R. Gerdes, H. Haak, R. W. Hallberg, W. Hazeleger, J. Jungclaus, W. G. Large, G. Madec, A. Pirani, B. L. Samuels, M. Scheinert, A. Sen Gupta, C. A. Severijns, H. L. Simmons, A. M. Treguier, M. Winton, S. Yeager and J. Yin, 2009: 
[revised manuscript text omitted]

---

## Referee Report (RR1)

Review of: A Last Glacial Maximum forcing dataset for ocean modelling

by Anne L. Morée and Jörg Schwinger

Manuscript number: essd-2019-79

The paper presents a dataset designed to compute boundary (surface) conditions for ocean general circulation model simulations of the last glacial maximum. The initial data comes from CMIP5/PMIP3 runs with coupled (ocean-atmosphere) general circulation models. The paper described the treatment made on original model outputs to derive the dataset.

This paper is well suited for the aims and scope of ESSD. The presentation is clear and fair, with the correct level of detail.

I have two main concern with this dataset. i) The rescaling method for temperature and humidity. ii) The actual interest of the dataset: the use of an average of CMIP5 ocean models to force ocean model might not be a relevant experiment protocol.

**Major concerns**

**Computation of values @10m ($t_{10m}$ and $q_{10m}$)**

The values at 2m height and the surface values are used to rescale the variables at 10m height. In atmospheric models, *t@2m* is not a prognostic variable. It is diagnosed with an iterative procedure (as in Large and Yeager, 2004). Input are surface values (temperature *tas*, pressure *psl*), and values at the first model level, generally between 10 to 100m height (temperature *temp[k=1]*, wind *u[k=1]*, *v[k=1]* and humidity *q[k=1]*). This computation of *t@2m* is an estimation, know to be very non precise, and not fully physically based. At least in the model it uses, or may use, the exact stability computed in the model, and the full high frequency outputs. Reapplying this procedure to recompute *t@10m* is prone to give large errors, especially because the estimated stability could be different from the one actually used in the model.

As the temperature of the first level is available in the CMIP5 database, it would be better to directly compute *t@10m* from *tas* and *temp[k=1]*. At least, the authors should check on one model than applying the procedure twice gives the same result than the direct computation.

• Which wind *u*, *v* and humidity *q* do you use for the computation ? It seems to be *u@10m*, *v@10m*, and *q@2m*, which for most model are also diagnostic variables computed with the same algorithm. This introduces an additional source of error.

• The procedure is applied on monthly mean. As the iterative procedure is highly non linear, is it justified ?

**Actual interest of mean values**

A procedure to force ocean models at the LGM is undoubtedly very useful. But I am very skeptical about using an average of different CMIP model outputs. The inter model spread

is very large, as mentioned in the paper. Atmospheric circulation pattern differs, and the internal coherence of a mean dataset is doubtful. It seems important that a user may evaluate the uncertainty coming from the forcing. And I can't imagine a way to build a variety of forcings from this dataset. The model spread is not relevant for this, as large values may locally and temporally come from different models. Perturbing the dataset with a fraction of the spread will generate incoherent patterns.

From the above rationale, the use of an inter model average is far from being an obvious protocol of LGM experiments. The dataset should probably include :

• Anomaly of individual models. For model that has a small ensemble, mean of ensemble could be provided if the intra ensemble spread is small (but how to define "small" ?).

• Absolute values for individual model, as applying anomalies to a given dataset is inconsistent. One may want to try the absolute values as forcing data.

**Minor concerns**

**Water budget closure**

The dataset provides precipitation anomalies. Evaporation will be computed from the CORE formula. To close the water budget, ocean modeller are missing the input from river and land ice melting. River input seems to be available for only 4 of the 5 model used in this study. Anyway, if the individual model data are given in the dataset (as suggested above), *friver* could be made available for some of them. This assume that a general interpolation procedure can be designed for all models, which maybe difficult because the variety of solution for river runoff in each model.

---

## Referee Report (RR2)

**REFEREE'S REPORT on Morée et al., essd-2019-79**

**Title: A Last Glacial Maximum forcing dataset for ocean modelling**

In this paper, Morée et al. present a new standard dataset for forcing of ocean-ice model simulations of the Last Glacial Maximum (LGM, ∼21ka BP). This kind of forcing dataset is already available for contemporary climate conditions but has so far been lacking for LGM climate. The availability of this type of dataset provides a possibility for standardized model intercomparison for the response of ocean-ice models to prescribed LGM conditions, and for moving from interglacial to glacial conditions. Such a dataset is also highly applicable when ocean-ice model simulations are preferable in the interest of reducing computational cost and run-times, as is clearly described in the paper. In previous reviews of this manuscript, reviewers raised major concerns about

1. the adequacy of using PMIP3 model averages, and about the fact that using such a model mean will be inconsistent with atmospheric dynamics,

2. re-referencing of 2 m to 10 m quantities,

3. the potential for using 3D model output, and

4. the lack of river runoff flux anomalies.

In the response to reviewers in the previous round, the authors have provided adequate responses to each of these concerns. First, they have clarified the manuscript according to their intentions for point 1. Next, they have re-evaluated points 2 and 3 and based on this evaluation decided to leave out the re-referencing step and found that the surface salinity anomaly can be improved by using 3D model output. For other fields, they explain why 3D data is not a viable/useful addition. Finally, for point 4, they have added the river runoff flux anomalies to their dataset. As these major concerns have been addressed, and I have no further major issues with the presented scientific approach, I recommend this manuscript for publication after minor revisions.

**General comments**

**Text:** The manuscript is well structured, and the text is generally easy to read, though some concepts and methods are introduced without proper explanation (see Specific comments). Therefore, it would be helpful if some clarifications were added. I have also given suggestions to smaller changes of the text that I find would improve the reading experience.

**Figures:** When showing anomaly figures, it is always easier for the reader to understand the fields resulting from these anomalies if the original fields (in this case, PMIP3 piControl) are also shown. I therefore suggest adding figures of the piControl fields for each of the variables to the supplementary material. I have also suggested to add model spread of the

sea-ice anomaly to supplementary material, as model spread in other variables is assumed to be based on the spread in modelled sea-ice.

**Data availability:** To simplify for the user, it would be great to add a download link which allows download of the entire dataset at once. I found it difficult to know if I had downloaded all files in the containers and sub-containers. Once downloaded, the dataset appears to be easy to handle and well documented, though I have not yet had the opportunity to apply it to simulations.

**Specific comments**

**Abstract**

Page 1, line 11: *"ocean-sea-ice"* – Though 'ocean-sea-ice' is adequate in the sense that the model does not simulate land-ice, it does not read well. I would suggest changing to 'ocean-and sea-ice'. The term ocean-sea-ice-atmosphere model is used later in the paper (P 2, L 8), and by clearly stating ocean and sea-ice here, the second contraction of these words becomes clearer.

Page 1, lines 12-14: *"The data presented here are derived from fully coupled paleoclimate simulations [...]"* – The previous two sentences (lines 8-12) seem to be about justifying the use of ocean-ice only models. Here, you very suddenly switch to talking about data from fully coupled simulations, which got me confused. Consider making this transition smoother by first introducing the problem addressed by your dataset (now on lines 21-22) rather than the dataset itself.

Page 1, line 19: *"pre-industrial times"* – The anomaly fields presented in this paper are based on PMIP3-simulations and are thus LGM-PI anomalies. However, it is also stated that the fields are optimized for use with the CORE forcing fields, which are not explicitly pre-industrial, but which have rather been evaluated against contemporary data. This should be clarified in the text (see also page 2, lines 32-33)

**1 Introduction**

Page 1, lines 26-28: *" [...] the LGM is extensively studied in modelling frameworks (Menviel et al., 2017; Brady et al., 2012; Otto-Bliesner et al., 2007)."* – For a topic that has been studied as extensively as the LGM in modelling frameworks, only citing three studies seems too little. I would suggest adding a few more relevant references and adding an e.g. before the given references.

Page 2, lines 3-5: *"Complex fully coupled models can typically not be run into full equilibrium (which requires hundreds to thousands of years of integration) due to computational*

*costs (Eyring et al., 2016). Therefore, the PMIP3 models exhibit model drift (especially in the deep ocean, e.g. Marzocchi and Jansen, 2017)."* – It would be valuable for the reader if you explained/exemplified, in one sentence, why equilibrium states are desirable and/or what kind of problems you get when analyzing a model that drifts.

Page 2, lines 8-9: *"We refer to a forced ocean model as a model of the ocean-sea-ice-atmosphere system in which the atmosphere is represented by prescribed 2-D forcing fields."* – It would be helpful to add a few references to widely used examples of such models, or papers that make intercomparisons.

Page 2, lines 22-27: starting with *"The description of the procedure [...]"* – I find it a little odd to present Section 3 before you present Section 2. In my opinion, it disturbs the fluidity in reading the text, and it is not clear to me why you have chosen to do so.

**2 General description of the dataset**

Page 2, line 29: *"pre-industrial state"* – What is the definition of pre-industrial in PMIP3?

Page 2, lines 32-33: *"and are optimized for use in combination with Coordinated Ocean-ice Reference Experiments (CORE) forcing fields (Griffies et al., 2009)."* – It should be clarified that these fields are in fact not strictly pre-industrial but rather corresponding to contemporary forcing. This was pointed out in a previous version of the manuscript, but unfortunately, that information has been removed.

Page 2, lines 33-34: *"The use of an anomaly forcing implies the assumption that no changes in temporal or spatial variability occurred between the lgm and piControl states beyond changes in the mean."* – What are the implications of this assumption? It does not seem to be discussed anywhere in the paper.

Page 3, lines 2-3: *"A discussion on the limitations of our dataset is provided in Sect. 4."* – Why not mention this in the introduction, where the rest of the structure of the paper is presented? It seems unnecessary that the reader needs to search for this information.

Page 3, lines 13-14: *"[...] proxy-based reconstructions are available for some of the variables (e.g., temperature)"* – 'some of' feels unnecessarily vague. You only have seven variables. You could say specifically for which variables proxy-based reconstructions are (currently) available and include relevant references.

Page 3, line 23: *"[...] which have been extensively used in the ocean modelling community (e.g. Griffies et al., 2009; Schwinger et al., 2016)."* – If they have been extensively used, it would be advisable to add a few more references here.

Page 3, lines 30-31: *"This choice ensures that the anomaly forcing data can be used with any pre-industrial land-sea mask."* – Is this true in any forced-model resolution? (I think here of models with lower-than-average horizontal resolution.)

**3 The variables**

Page 4, line 4: *"[...] a 6-hour time resolution"* – There is no description in the text of why some variables are time interpolated to 6-hour time resolution and some to a daily time resolution. It would be helpful for the reader if this was outlined in Section 2.

Page 4, line 8: *"without any strong spatial pattern"* – Here, I disagree with the authors. I do see a clear spatial pattern, with more spread in the Northern Hemisphere and particularly in the western boundary current regions/close to the major Northern ice sheets. I would therefore like to know what the authors base this statement on.

Page 5, lines 18-20: *"The inter-model spread ($\sim$1-3 m s$^{-1}$) has little structure except for the $\sim$4-5 m s$^{-1}$ disagreement in the Southern Ocean south of $\sim 40°S$, and the $\sim$3-5 m s$^{-1}$ disagreement in the North Atlantic (Fig. 6)."* – These disagreement zones are indeed quite pronounced. It gets me wondering what the likely explanation for the disagreements in each of these two zones is. Later, in the Discussion (page 8, lines 2-3), you state that these explanations are beyond the scope of this study, though in this section you do provide explanations for some of the other variables. I would argue that if these explanations have been provided in other PMIP3-studies, it would not be a major effort for the authors to include them. For the North Atlantic, this is not evident, but for the Southern Ocean, is it possible that the explanation is found in the differences in the jet position relative to the sea-ice edge, as described by Sime et al., 2016? (Sime, L. C., Hodgson, D., Bracegirdle, T. J., Allen, C., Perren, B., Roberts, S., de Boer, A. M., 2016: Sea ice led to poleward-shifted winds at the Last Glacial Maximum: the influence of state dependency on CMIP5 and PMIP3 models, Clim. Past, 12, 22412253, doi: /10.5194/cp-12-2241-2016)

Page 6, lines 24-26: *"Regarding the dynamical inconsistencies, it is important to note that the CORE forcing itself is a mixture of reanalysis and 25 observational data products and as such not dynamically consistent."* – This sentence is phrased in a way that makes me feel like the CORE forcing has been previously discussed in the paragraph, which is not the case. Consider rephrasing or adding a sentence 'Our dataset has been adapted to be an extension of the CORE forcing.' (or similar) before this sentence.

Page 8, lines 2-3: *"The attribution of the model spread to specific processes is beyond the scope of this article [...]"* – Yet the authors do attempt to provide these explanations for several variables. For a user of the dataset, I think it would be useful to know where the model spread originates from, and while reading the article, I found it somewhat frustrating that some of these explanations were missing (see my comment for page 5, lines 18-20). I

feel that, in those cases where that information is available from other PMIP3 research, it is something that the authors could provide without additional analysis. For those variables that explanations for the model spread are given, there are generally no motivations for these explanations, for example in the form of a cited paper or a confirming supplementary figure (see e.g. page 4, lines 14-15; and page 5, lines 25-26, which could both be confirmed with a figure of model spread in sea-ice cover). Based on the fact that some explanations are indeed already given, I would advise to rephrase this sentence to 'The complete attribution of the model spread [...]' , whether or not any further explanations can be added.

Page 8, line 5: *"Finally, there is no other way [...]"* – I would suggest expressing this a bit more cautiously, by changing to Finally, there is currently no other way [...]

**Technical corrections**

Page 3, line 9: *"yearly"* – annual

Page 3, line 19: *"The SSS fields is [...]"* – The SSS fields are [...]

Page 3, line 35: *"under the assumption of an unchanged spatial and temporal variability"* – remove an

Page 4, lines 6 and 8: *"2-3 kg $kg^{-1}$" and "1-2 kg $kg^{-1}$"* – According to the colourbar in Fig. 1, the magnitude of the specific humidity anomaly and spread is 10-3 kg $kg^{-1}$.

Page 4, line 23: *"symmetrically"* – symmetrical, i.e. a symmetrical spatial pattern, though it is not entirely clear what symmetrically refers back to here. Consider either changing to symmetrical or rephrasing the sentence.

Page 5, line 24: The temperature anomalies of 10K and ∼2.5K are presumably supposed to be negative.

---

## Editor Decision (ED1)

Dear Anne and coauthors,

first of all, I acknowledge the delay for this manuscript dues to the finalization of the PhD thesis of the lead author and proceeds with this manuscript in February 2020. I hope that your defense of the thesis was successful and that you find more time for this manuscript now.

I am summarizing below the agreements between you (the authors) and the TE (Kirsten Elger) for this manuscript (based on several email exchanges and a skype call in late November 2019):

Agreed Workflow suggested by the authors:

- We write a public response to reviewer #3 (So that the interactive discussion would show a RC3 and AC3);
- More reviewers can then review the current version of the manuscript (suggestions below*);
- Then we would, if necessary, make a new version of the manuscript/dataset after the reviews (=reviewer #3 + some more).
- Let us know whether this would work for you. We would then need the Copernicus system to show the RC3 and get a possibility to upload a AC3.

Detailed answers to specific requests below. Please proceed with the revision of this manuscript at your earliest convenience.

TE:

All three reviewers request a significant change of your data and paper. There is a general concern about the very little number of input variable (due to the use of the surface variables and not the 3D fields of CMIP5/PMIP3 data for which there would be many more data available - this is the main question of Reviewer #2).

I have also contacted Reviewer #1 and asked whether a new setup of your model would be possibly improving your data quality. The answer here was "It's possible the group would be willing to include more models and variables, though it could be a significant time investment, hard for me to say. My guess would be that their results would change significantly, which is one of the reasons I'm skeptical about this endeavor. The CORE2 forcings have been a powerful tool because they combine different kinds of information and converge on a reasonable set of atmospheric forcings; if more observations were added, the results probably wouldn't change too much. At the LGM, there is no guarantee that averaging models will converge on anything at all, let alone the "right" annual cycle. So while there's value in what they're trying to do, estimating the LGM annual cycle and its uncertainty is a deep question and I worry that characterizing the present treatment as an LGM CORE2, even with more models added, risks oversimplifying the problem."

Authors comments:

(1) Use of multi-model mean fields

Regarding Reviewer #1, and the comment on the robustness of the dataset: We show that after adding the GISS model (difference between version 1 and version 2 of the dataset), the

mean anomaly fields look almost identical (i.e., the figures in v2 of the manuscript are nearly the same as in v1), showing that adding more data will likely not significantly change our results. Regarding Reviewer #3, providing individual anomalies for each of the models is something we could certainly provide.

Regarding the comments of reviewer #1, we stress (again) that we do not try to provide a true estimate of the LGM annual cycle, nor are we able to provide a LGM "normal year" forcing – this is because available model output or other data is too limited, not because we are reluctant to take into account suggestions to improve our data set. As described in our author response, we aim to provide a large scale monthly mean anomaly field that can be used with any type of pre-industrial ocean forcing (but formatted for easy use with CORE forcing, see also the first paragraph of Sect. 2), to obtain an estimate of the LGM atmospheric state. We believe that this is currently (given the available data) the best estimate we can produce. Regarding the comments on averaging over multiple models (Reviewer #1 and #3), this is an accepted practice in model intercomparison studies (e.g. HappiMIP uses multi model average SSTs, Mitchell et al. (2017)) and to force stand-alone models (Muglia et al., 2015/2018), as well as outside of the paleoclimate and MIP community (f.e., Chowdhury and Behera, 2019). There are hundreds of papers analyzing the multi model mean and spread of CMIP5 models. We have discussed and defended the inconsistencies that will occur through such averaging in our earlier author response and incorporated that in the main text (Sect. 2). Nevertheless, as mentioned earlier, we could include individual model anomalies in an updated version of our data set.

(2) Use of 3D model output and calculation of 10m temperature and humidity

Reviewer #2 shared some of his/her remaining concerns with you, as far as we understand after reading our author response. We understand this is mainly about the use of the 2D surface fields. First, we would like to note that the use of 3D output would actually not lead to many more available data, as it would be just seven models (adding CCSM and MPI-ESM) as compared to 5 models in version 2 of our dataset.

Second, unfortunately, we will not be able to use 3D model output, because the consistent calculation of 10m temperature and humidity would require the implementation of parts of the boundary layer scheme for each model, which is beyond what we could do. But, more importantly, we cannot agree with the reviewer (#3) that such a procedure would result in more accurate 10 metre temperature and humidity after taking into account limitations due to data availability. After receiving the reviews we re-discussed this issue with an expert in boundary layer meteorology. The bottom line is that an inconsistency with the original model is unavoidable as the calculation is done off-line (and due to data availability: only climatological monthly means are available in PMIP), but the error made will be generally much smaller by using our method (shifting from 2 to 10 meters with an inconsistent method) than doing what the reviewers ask for (shifting from the lowest 3D model level, i.e. over a factor of 10 larger vertical distance, with a still inconsistent method as it is done off-line and on climatological monthly mean data). We would be able to address the point of reviewer #3 about estimating the impact of using monthly averages instead of higher resolution data in our calculations.

(3) Inclusion of river input

Regarding river fluxes (variable 'friver'), an anomaly field would not be meaningful due to differences in the land-sea mask. Nevertheless, we see an opportunity to provide the user with (basin-scale) LGM-PI changes in river runoff based on the different models, which we expect will satisfy the reviewer.

TE: Additional comment:

Seeing the third review (who reviewed already the revised version) also mentioning major concerns with the rescaling method for temperature and humidity and the gerenal interest of the dataset as "the use of an average of CMIP5 ocean models to force ocean model might not be a relevant experiment protocol".

Authors: this will be addressed in a direct answer to Reviewer #3 in the public discussion

Many thanks and best regards,

Kirsten Elger

---

## Author Response (AR2)

7th of May, 2020

Dear Editor Kirsten Elger,

Thank you for your time to handle our manuscript. We have written a response to the comments of reviewer #3 below. Additionally, we included a general reply to some of the main points raised by the three reviewers (where reviewers #1 and #2 have reviewed the first version of the manuscript, and reviewer #3 has reviewed the second version of the manuscript) to further elaborate on our choices made for the third version of the manuscript. To address the comments of the different reviewers, we included a new section on the limitations and robustness of the dataset at the end of the manuscript, as well as a new figure (A1) and smaller changes throughout the text. There are revisions that we potentially could make to the data set (including basin wide river runoff anomalies and omitting the re-referencing procedure). Since we already have two data set versions archived with a doi, we refrained from preparing a new data set version at this time as discussed with you, and would like to await the further outcome of the review process.

Yours sincerely,
Anne Morée and Jörg Schwinger

**General reply to some of the main points raised by reviewers**

Since some time has passed since our first submission, we would like to start this response letter with some general considerations, taking into account the most important points raised by the three reviewers of our manuscript. We believe that there has been some misconception of what we intend to provide with our data set and what is needed and is practical in forced ocean modelling.

1) We do not aim to provide an LGM normal year forcing (as available data do not allow this). It is our *intention* to provide large scale, LGM-PI anomaly fields derived from PMIP3 models. This is not (only) due to data limitations, but, equally important, due to the fact that individual model results are highly uncertain, particularly at smaller spatial scales. Calculating estimates of large scale anomalies as a model mean is an established practise in climate research. One major point raised by the reviewers is that a model mean will be inconsistent with atmospheric dynamics (i.e. the underlying physical equations of motion/state will not be satisfied by the resulting forcing fields). This is correct, but not particularly relevant in the context of forced ocean modelling. The CORE forcing itself is constructed based on the NCEP-R1 reanalysis, to which either observation based corrections are applied (temperature, winds) or, for some variables (radiation and precipitation fields), the reanalysis data is replaced by observational estimates. No efforts are made to preserve dynamical consistency. This CORE forcing data set is widely used (lately in the CMIP6 endorsed OMIP model intercomparison), and we do not see a reason why something that is acceptable for CMIP6 should not be acceptable for our data set. Also, the use of model mean fields as boundary condition in modelling is not unheard of. E.g., HappiMIP (Mitchell et al., 2017) uses multi model average SSTs as forcing. Also outside of the paleoclimate and MIP community multi-model CMIP output is used as forcing (e.g., Chowdhury and Behera, 2019). Last, forcing

specifically with PMIP3 anomalies (only for fewer variables) is practiced in the state of the art studies by for example Muglia et al. (2015/2018) and Khatiwala et al. (2019).

The use of individual model anomalies (or absolute values) would be of little practical value, we believe (although we could easily provide these). First, this is an issue of computational resources. Very long integration times are necessary to run a forced ocean model into equilibrium with LGM forcing, particularly for biogeochemistry in general and carbon isotopes in particular. Using five model integrations based on five individual PMIP model forcings is not really an option. Even more important though, the ocean model's salinity restoring scheme has to be tuned (as described in our manuscript) for each individual forcing to produce a reasonable large scale ocean circulation. We do not believe that any ocean modelling group would want to apply this time consuming procedure for 5 or even more individual forcings.

2) Re-referencing of 2m to 10m quantities: This is a procedure that is applied to NCEP 2m temperature and specific humidity when the NCAR reanalysis data is processed for the CORE forcing. We have  applied the same procedure to PMIP3 model output for the first versions of our data set. Reviewers have criticized this as a) inconsistent with the estimated boundary layer stability and the algorithms used in the models from which the input quantities are taken, and b) that using monthly averages as an input for this procedure is introducing errors. Both are valid points, and we have done some further analysis on point b), which is presented in detail below (see our specific response to reviewer #3, comment 1). We summarize the results here: Over the open ocean re-referencing only has a small effect (the difference between 2m and 10m temperature is less than 0.1 K for the vast majority of ocean grid points, Fig. 1 columns 1 and 3). The effect is larger over sea ice (typically around 1 to 1.5 degree) and land. The error made by using monthly mean inputs for the re-referencing is also small over the open ocean, but it can be substantial over sea-ice (and over land) (Fig. 1 column 2). Still, these errors are small compared to the uncertainty of the underlying model ensemble, which is larger than 8 K almost everywhere north and south of 60° (ensemble range of PMIP3 models, Fig. 1 last column). Results for specific humidity are discussed below (see our specific response to reviewer #3, comment 1).

Given this analysis, we conclude that it would be more robust to skip the re-referencing step altogether. For our data set we are taking the difference between two temperatures, so the effect of omitting the re-referencing is virtually zero over the open ocean (<0.1 K for the vast majority of open ocean grid cells; Fig 1 column 3). Larger differences occur over sea ice, particularly over the central Arctic Ocean. Here, the 2m temperature anomaly is up to 4 K larger (colder) than the anomaly based on 10m values. In practise however, these differences at very high latitudes will have little influence on model simulations forced by our data set, since the ocean is anyway insulated by thick sea-ice from the atmosphere. Also, the PMIP model ensemble range exceeds 20 K over the central Arctic Ocean.

Given the lack of time resolved input data, the unavoidable error made by using time-average input to re-referencing, and given the small impact on the actual anomaly fields (relative to the uncertainties), we propose to skip the re-referencing step in a revised version of our data set, and discuss the impact of this in our revised manuscript (in a new section, Sect. 4).

3) Using 3d model output

It has also been brought up that we could use more models if we would use 3D model output. In our revised data set, we have improved the surface salinity anomaly estimate by using 3D output (increasing number of available models from 2 to 5). It is also true that we could use 7 instead of 5 models for atmospheric conditions if using 3D output. However, we would need to calculate 10m (or 2m) temperature and specific humidity from the original 3D fields. Given our analysis on the errors made by re-referencing using monthly mean fields, this is not an option. We would like to stress that the calculated model mean anomalies are already quite robust with 5 models, i.e. the addition of one model from 4 to 5 models did not change the results significantly, as visible from the difference between version 1 and 2 of our dataset.

4) Freshwater budget

Our data set was criticized for not including river runoff flux anomalies, leaving out one component of the water cycle. This is correct, and we would be able to amend this in a third version of our data set. We propose providing basin wide total and fractional anomalies, which could be used to scale the pre-industrial river fluxes by modelling groups. We note that providing gridded anomalies for river runoff is not an option because of different land-sea masks and river mouth locations in the different models.

**Author Comment to Review #3**

We thank reviewer #3 for his/her time to provide constructive feedback on the version of our manuscript from 10th of September 2019. Our response to the three comments is provided below. Specifically, we propose to add data on basin scale river-runoff anomalies to our data set and to omit the re-referencing procedure, based on an analysis presented below. We also include a new section (Sect. 4) on the robustness and limitations of the dataset and a new Figure (A1) to address the reviewer's concerns.

Yours sincerely,
Anne Morée and Jörg Schwinger

1. **Computation of values @10m (t10m and q10m)**

The values at 2m height and the surface values are used to rescale the variables at 10m height. In atmospheric models, t@2m is not a prognostic variable. It is diagnosed with an iterative procedure (as in Large and Yeager, 2004). Input are surface values (temperature tas, pressure psl), and values at the first model level, generally between 10 to 100m height (temperature temp[k=1], wind u[k=1], v[k=1] and humidity q[k=1]). This computation of t@2m is an estimation, know to be very non precise, and not fully physically based. At least in the model it uses, or may use, the exact stability computed in the model, and the full high frequency outputs. Reapplying this procedure to recompute t@10m is prone to give large errors, especially because the estimated stability could be different from the one actually used in the model. As the temperature of the first level is available in the CMIP5 database, it would be better to directly compute t@10m from tas and temp[k=1]. At least, the authors should check on one model than applying the procedure twice gives the same result than the direct computation.

● Which wind u, v and humidity q do you use for the computation? It seems to be u@10m, v@10m, and q@2m, which for most model are also diagnostic variables computed with the same algorithm. This introduces an additional source of error.
● The procedure is applied on monthly mean. As the iterative procedure is highly non linear, is it justified?

*Author response to comment 1*

*We have analysed the error made by using monthly means as an input to the re-referencing (the 2nd bullet point in the comment above). For this exercise we took the original NCAR reanalysis output (6-hourly time resolution), formed monthly means and applied the re-referencing. We compared this output with the correct re-referencing (i.e., 6-hourly output re-referenced and then averaged). We call the difference between the two the "re-referencing error" in the following text. We note that we can only derive this error for the pre-industrial state due to data availability, while our data set relies on the difference between LGM and PI quantities. If the re-referencing would have the same effect (and error) under an LGM and PI state, we could omit this procedure for creating our data set. Below we show that (over the ocean), the effect of re-referencing is small, except for sea ice covered regions. Consequently, anomalies based on 2m or 10m quantities are*

*virtually identical, again except for ice covered regions. Results of these analyses are shown in Fig. 1 for temperature, and Fig. 2 for specific humidity.*

*For temperature, re-referencing is mainly important over sea ice (disregarding the land, which is not relevant for our ocean forcing data set), where T(10m) is typically 1 to 1.5 K warmer than T(2m) (Fig 1, column 1). For the majority of open ocean grid cells the effect of re-referencing is smaller than 0.1 K. The re-referencing error is also small (<0.1 K) over the open ocean, but can be substantial over sea ice (Fig. 1 column 2).*

*For specific humidity the role of sea ice is less pronounced. The effect of re-referencing is largest over the low latitude ocean (between 40°S and 40°N) due to high absolute humidity values (Fig. 2 column 1). The re-referencing error is largest in the subtropics and again over sea ice (Fig. 2 column 2). Particularly at high latitudes the re-referencing error can be larger than the effect from the re-referencing itself (albeit for small absolute values).*

*Since we are actually interested in the difference of two temperatures for our dataset, we analyze next the effect of taking the 2m temperature (and specific humidity) anomaly without any re-referencing compared to taking the 10m anomalies. This is shown in Figs. 1 and 2, 3rd column. Consistent with our analysis above, for both temperature and specific humidity, the difference is generally small (<0.1 K and <0.05 g/kg, respectively) over the open ocean. For specific humidity, we find differences that are of the same order of magnitude as the re-referencing error over sea ice. For temperature the difference is up to 4 K over the central Arctic Ocean. This is most probably caused by a more stable boundary layer and thus a larger effect of re-referencing under LGM compared to PI conditions. Also the re-referencing error could be smaller if the LGM boundary layer over sea ice is more stable (i.e., less alteration to an unstable state justifies usage of monthly means better), but this is difficult to quantify with available data.*

*We come to the conclusion that it might be better to omit the re-referencing step. The re-referencing of monthly mean values comes with a significant error that will be different between PI and LGM states for very stable boundary layers (over sea ice). The omission of re-referencing has no impact over open ocean regions. For Arctic temperatures, there seems to be a systematic difference between 2m and 10m temperature anomalies of up to 4 K. We note that this difference will not significantly influence simulation results, since at these latitudes the ocean is covered by thick sea ice in the LGM anyway. Also, we note that the large PMIP3 model spread at high latitudes (Fig 1 and 2, last column) might serve to justify this decision.*

*We therefore propose to omit the re-referencing step for a revised version of our data set. We have amended our manuscript to discuss the assumptions and limitations of our data set, and we have included a discussion of omitting the re-referencing step (new Sect. 4).*

**2. Actual interest of mean values**

A procedure to force ocean models at the LGM is undoubtedly very useful. But I am very skeptical about using an average of different CMIP model outputs. The inter model spread is very large, as mentioned in the paper. Atmospheric circulation pattern differs, and the internal coherence of a

mean dataset is doubtful. It seems important that a user may evaluate the uncertainty coming from the forcing. And I can't imagine a way to build a variety of forcings from this dataset. The model spread is not relevant for this, as large values may locally and temporally come from different models. Perturbing the dataset with a fraction of the spread will generate incoherent patterns. From the above rationale, the use of an inter model average is far from being an obvious protocol of LGM experiments. The dataset should probably include:

- Anomaly of individual models. For model that has a small ensemble, mean of ensemble could be provided if the intra ensemble spread is small (but how to define "small" ?).
- Absolute values for individual model, as applying anomalies to a given dataset is inconsistent. One may want to try the absolute values as forcing data.

*Author response to comment 2*
*We provide individual model anomalies for each variable in a new figure A1 to visualize the difference between the model anomalies for each variable in the dataset as to inform the reader with more detail than just the model spread in the dataset. As described in our general response, we would also be able to provide such individual model anomalies and/or absolute fields in a revised version of our data set. As outlined however (see above, point 1), we believe that the individual model fields would be of little practical value. This approach is limited by computational resources and trade-offs between integration length, ensemble size, and the need of tuning for each individual forcing. We believe that the use of an ensemble mean anomaly forcing with typically long paleo-simulation integration times is a valid and useful application. Regarding the missing "internal coherence of a mean dataset", we refer to our general response above (point 1).*

**3. Water budget closure**
The dataset provides precipitation anomalies. Evaporation will be computed from the CORE formula. To close the water budget, ocean modeller are missing the input from river and land ice melting. River input seems to be available for only 4 of the 5 model used in this study. Anyway, if the individual model data are given in the dataset (as suggested above), friver could be made available for some of them. This assume that a general interpolation procedure can be designed for all models, which maybe difficult because the variety of solution for river runoff in each model.

*Author response to comment 3*
*We propose to add information on river runoff anomalies to a revised version of our manuscript. Since a gridded anomaly field would not be practical due to differences in the land-sea mask and river mouth locations between the models (as noted by the reviewer), we propose to calculate river runoff-anomalies (absolute values and fractional change) on basin scale (North/South Atlantic, North/South Pacific, and Indian Ocean). These anomalies could then be used by modelling groups to scale their pre-industrial river runoff.*
*We note that forced ocean models will inevitably have an imbalance between freshwater sources and sinks (there is no regulating feedback in such a model setup). For longer integrations (several hundred years or longer) such models usually implement a balancing of freshwater fluxes to avoid long term salinity drift. Such balancing can be accomplished e.g. by increasing/decreasing the prescribed precipitation fluxes based on diagnosed imbalances.*

Last, we noticed that the wind anomaly was not updated in our revision of Fig. 5, which we corrected now.

Yours sincerely,
Anne Morée and Jörg Schwinger

[Figure]

**Figure 1:** (1st column) climatological difference between T(2m) and (after re-referencing using 6-hourly input data) T(10m); (2nd column) re-referencing error defined as the difference between T'(10m), which is calculated using monthly mean input data (i.e. as done in our dataset), and T(10m); (3rd column) difference between the LGM-PI anomaly based on T(2m) and T(10m), where 10m temperatures for LGM and PI have been re-referenced using climatological monthly mean PMIP3 output; (4th column) uncertainty estimate (ensemble range of the 5 PMIP3 models as provided in our dataset) of our LGM-PI temperature anomaly field. Masked grid cells in the first three columns values smaller than 0.1 K. Climatologies of the 1st and 2nd columns are calculated over 30 years of NCEP-R1 data (1980-2009).

[Figure]

**Figure 2:** as Fig. 1 but for specific humidity. Masked grid cells in the first three columns indicate values smaller than 0.05 g/kg.

[revised manuscript text omitted]

---

## Author Response (AR3)

Dear reviewer, dear Editor Kirsten Elger,

30 September 2020

Thank you for your time and efforts to provide us with this constructive and clear review of our manuscript version from May 7, 2020. Our reply to each of your comments is provided below (in *italics*).

The new version of the dataset, version 3, is available at https://doi.org/10.11582/2020.00052. We updated the figures in the manuscript accordingly. As the only difference with dataset version 2 is the re-referencing of humidity and temperature, only Figure 1 and 7 (anomaly and multi-model mean spread for humidity and temperature) and the tas and huss variables in Fig. A1 are slightly different. Note as well that we included the river runoff anomalies in a new Table 4, and a description thereof in the section on precipitation (Sect. 3.4).

Yours sincerely,

Anne Morée and Jörg Schwinger
* * *
**General comments**

**Text**: The manuscript is well structured, and the text is generally easy to read, though some concepts and methods are introduced without proper explanation (see Specific comments). Therefore, it would be helpful if some clarifications were added. I have also given suggestions to smaller changes of the text that I find would improve the reading experience.

*Thank you for providing constructive input to improve the text - see our reply below for details on how we addressed your concerns.*

**Figures**: When showing anomaly figures, it is always easier for the reader to understand the fields resulting from these anomalies if the original fields (in this case, PMIP3 piControl) are also shown. I therefore suggest adding figures of the piControl fields for each of the variables to the supplementary material. I have also suggested to add model spread of the sea-ice anomaly to supplementary material, as model spread in other variables is assumed to be based on the spread in modelled sea-ice.

*Thank you for this suggestion. We decided to add the piControl multi-model annual mean fields for all eight variables by adding a new appendix figure A2 (as an appendix for easier reference than a separate supplementary material file). The new Fig. A2 is introduced in Sect. 2 directly after introducing Fig. A1.*

*We visualize the model spread of the sea ice cover anomaly in a new figure A3 (based on variable 'sic' in CMIP/PMIP, % fraction of grid cell covered by sea ice) and refer to this new figure where appropriate in Sect. 3 (see more details in our reply below).*

**Data availability**: To simplify for the user, it would be great to add a download link which allows download of the entire dataset at once. I found it difficult to know if I had downloaded all files in the containers and sub-containers. Once downloaded, the dataset appears to be easy to handle and well documented, though I have not yet had the opportunity to apply it to simulations.

*Download of all data with one click would indeed be a good improvement. We inquired with the NIRD Research Data Archive whether this is a possibility. They suggested including a zipped archive of all files as an additional file, which we have done and named 'Moree2020v3.tar.gz'.*

**Specific comments**
**Abstract**
Page 1, line 11: "ocean-sea-ice" – Though 'ocean-sea-ice' is adequate in the sense that the model does not simulate land-ice, it does not read well. I would suggest changing to 'ocean and sea-ice'. The term ocean-sea-ice-atmosphere model is used later in the paper (P 2, L 8), and by clearly stating ocean and sea-ice here, the second contraction of these words becomes clearer.
*Thank you for this suggestion, we changed the manuscript accordingly.*

Page 1, lines 12-14: "The data presented here are derived from fully coupled paleoclimate simulations [...]" – The previous two sentences (lines 8-12) seem to be about justifying the use of ocean-ice only models. Here, you very suddenly switch to talking about data from fully coupled simulations, which got me confused. Consider making this transition smoother by first introducing the problem addressed by your dataset (now on lines 21-22) rather than the dataset itself.
*Thank you for noting this. We restructured the abstract by moving back and rephrasing the sentence that introduced the problem addressed by our dataset (lines 21-22) such that it is placed before lines 12-14 - where we state that we base our dataset on fully coupled models.*

Page 1, line 19: "pre-industrial times" – The anomaly fields presented in this paper are based on PMIP3-simulations and are thus LGM-PI anomalies. However, it is also stated that the fields are optimized for use with the CORE forcing fields, which are not explicitly pre-industrial, but which have rather been evaluated against contemporary data. This should be clarified in the text (see also page 2, lines 32-33)
*We replaced the word 'times' with 'simulations' and explained the definition of the piControl simulation in Sect. 2. To clarify that the CORE is not strictly a pre-industrial forcing, we added a sentence in Sect. 2: 'We note that the CORE forcing is based on modern era (1948-2009) reanalysis and observations, and thus is not a pre-industrial forcing. However, the anthropogenic climate signal contained in these data is relatively small, particularly in comparison to the uncertainties of the LGM-PI anomalies (see below).'*

**Introduction**
Page 1, lines 26-28: " [...] the LGM is extensively studied in modelling frameworks (Menviel et al., 2017; Brady et al., 2012; Otto-Bliesner et al., 2007)." – For a topic that has been studied as extensively as the LGM in modelling frameworks, only citing three studies seems too little. I would suggest adding a few more relevant references and adding an e.g. before the given references.
*There are indeed many other studies we could have cited here, and an 'e.g.' is therefore in place. We added 'e.g.' and included several more (ocean) modelling studies in our citation.*

Page 2, lines 3-5: "Complex fully coupled models can typically not be run into full equilibrium (which requires hundreds to thousands of years of integration) due to computational costs (Eyring et al., 2016). Therefore, the PMIP3 models exhibit model drift (especially in the deep ocean, e.g.

Marzocchi and Jansen, 2017)." – It would be valuable for the reader if you explained/exemplified, in one sentence, why equilibrium states are desirable and/or what kind of problems you get when analyzing a model that drifts.

*We added the sentence 'Since significant differences between a (drifting) non-equilibrated state and the equilibrium model state can impede comparison of model results with proxy data, a well equilibrated model with minimal drift is desirable.'*

Page 2, lines 8-9: "We refer to a forced ocean model as a model of the ocean-sea-iceatmosphere system in which the atmosphere is represented by prescribed 2-D forcing fields." – It would be helpful to add a few references to widely used examples of such models, or papers that make intercomparisons.

*We followed this suggestion by adding the sentence 'Such model set-ups have been extensively used in model intercomparison studies such as the Coordinated Ocean-ice Reference Experiments (COREs; Griffies et al., 2009), and more recently in the CMIP6 Ocean Model Intercomparison Project (OMIP; Griffies et al. 2016).*

Page 2, lines 22-27: starting with "The description of the procedure [...]" – I find it a little odd to present Section 3 before you present Section 2. In my opinion, it disturbs the fluidity in reading the text, and it is not clear to me why you have chosen to do so.

*We agree with the reviewer that the text has a better flow when Sect. 2 is introduced before Sect. 3, and we moved the sentence that introduces Sect. 2 forward in order to do so.*

**2 General description of the dataset**

Page 2, line 29: "pre-industrial state" – What is the definition of pre-industrial in PMIP3?

*The piControl experiments which represent 'the pre-industrial state' are the same as in CMIP5 and represent the year 1850. We clarified this in the text.*

Page 2, lines 32-33: "and are optimized for use in combination with Coordinated Oceanice Reference Experiments (CORE) forcing fields (Griffies et al., 2009)." – It should be clarified that these fields are in fact not strictly pre-industrial but rather corresponding to contemporary forcing. This was pointed out in a previous version of the manuscript, but unfortunately, that information has been removed.

*We clarified this by adding the sentence 'We note that the CORE forcing is based on modern era (1948-2009) reanalysis and observations, and thus is not a pre-industrial forcing. However, the anthropogenic climate signal contained in these data is relatively small, particularly in comparison to the uncertainties of the anomaly fields (see below).'*

Page 2, lines 33-34: "The use of an anomaly forcing implies the assumption that no changes in temporal or spatial variability occurred between the lgm and piControl states beyond changes in the mean." – What are the implications of this assumption? It does not seem to be discussed anywhere in the paper.

*We have removed this sentence, since this problem is already discussed in Section 4 (limitations of our dataset), and since we feel that this sentence is a bit out of context in Section 2. In Section*

*4, we have added a brief discussion of the implications: 'We can currently not quantify the implications of this assumption, but future phases of PMIP (providing simulation output with higher temporal resolution) might alleviate this problem.'*

Page 3, lines 2-3: "A discussion on the limitations of our dataset is provided in Sect. 4." – Why not mention this in the introduction, where the rest of the structure of the paper is presented? It seems unnecessary that the reader needs to search for this information.
*We agree with the reviewer that Sect. 4 could be introduced earlier in the paper, and added the sentence 'Limitations of the dataset are discussed in Sect. 4.' to the structure description at the end of Sect. 1.*

Page 3, lines 13-14: "[...] proxy-based reconstructions are available for some of the variables (e.g., temperature)" – 'some of' feels unnecessarily vague. You only have seven variables. You could say specifically for which variables proxy-based reconstructions are (currently) available and include relevant references.
*We are not aware of a purely proxy-based reconstruction with global coverage for any of the variables, which would be required for the creation of a proxy-based model forcing. Nevertheless, in this sentence we mean to refer to the availability of some regional/local information for some of the variables. Combination of such proxy data with output from (PMIP) model runs can create global coverage for air-temperatures (e.g., Annan and Hargreaves, 2015). Similarly, the development of new methods to reconstruct humidity (e.g., Alexandre et al., 2018) or precipitation (e.g., Mendes et al., 2019) shows promise for these variables as well, although the proxy data are only available over the continents. Large-scale proxy-based information about wind speed and direction (e.g., Markewich et al., 2015) is mostly qualitative. Information on incoming ('downwelling') radiation can be reconstructed based on the Milankovitch parameters for the top of the atmosphere, but no proxy exists which could quantitatively reconstruct the near-surface radiation which would be affected by cloud cover, humidity, etc. Last, sea surface salinity is (like precipitation) only locally reconstructed based on proxies and highly uncertain (Rohling, 2000).*
*In summary, the variables we present in our dataset are generally only reconstructed using proxies over land and in qualitative form, and are not available with global coverage without involving model data. We will summarize the above by replacing this sentence with a more elaborate paragraph covering the above.*

Page 3, line 23: "[...] which have been extensively used in the ocean modelling community (e.g. Griffies et al., 2009; Schwinger et al., 2016)." – If they have been extensively used, it would be advisable to add a few more references here.
*We wish to highlight here that they are common because they are a standard in model intercomparison studies. We added a few more references and specifically refer to the fact that CMIP6 OMIP experiments rely on CORE format forcing and include the relevant CMIP references.*

Page 3, lines 30-31: "This choice ensures that the anomaly forcing data can be used with any pre-industrial land-sea mask." – Is this true in any forced-model resolution? (I think here of models with lower-than-average horizontal resolution.)

*We indeed expect this to be true for any pre-industrial land mask, as we extended the WOA coastline 1.5 degrees inland. However, the statement may be too strong (for very low resolution models) and we rephrased this sentence and added 'likely' to it.*

**3 The variables**
Page 4, line 4: "[...] a 6-hour time resolution" – There is no description in the text of why some variables are time interpolated to 6-hour time resolution and some to a daily time resolution. It would be helpful for the reader if this was outlined in Section 2.
*We chose to time-interpolate the variables to their respective time resolution in the CORE Normal Year Forcing format (CORE-NYF; Large and Yeager, 2004). This sentence is now added to Sect. 2.*

Page 4, line 8: "without any strong spatial pattern" – Here, I disagree with the authors. I do see a clear spatial pattern, with more spread in the Northern Hemisphere and particularly in the western boundary current regions/close to the major Northern ice sheets. I would therefore like to know what the authors base this statement on.
*We agree that this statement was not accurate. We changed the text as follows: 'The model spread of the anomaly shows a disagreement between the PMIP3 models generally in the order of $1$-$2 \times 10^{-3}$ kg kg$^{-1}$, but is larger (up $4 \times 10^{-3}$ kg kg$^{-1}$) in the northern hemisphere western boundary current regions and close to the Arctic ice edge.'*

Page 5, lines 18-20: "The inter-model spread ($\sim$1-3 m s$^{-1}$) has little structure except for the $\sim$4-5 m s$^{-1}$ disagreement in the Southern Ocean south of $\sim 40 \circ$S, and the $\sim$3-5 m s$^{-1}$ disagreement in the North Atlantic (Fig. 6)." – These disagreement zones are indeed quite pronounced. It gets me wondering what the likely explanation for the disagreements in each of these two zones is. Later, in the Discussion (page 8, lines 2-3), you state that these explanations are beyond the scope of this study, though in this section you do provide explanations for some of the other variables. I would argue that if these explanations have been provided in other PMIP3-studies, it would not be a major effort for the authors to include them. For the North Atlantic, this is not evident, but for the Southern Ocean, is it possible that the explanation is found in the differences in the jet position relative to the sea-ice edge, as described by Sime et al., 2016? (Sime, L. C., Hodgson, D., Bracegirdle, T. J., Allen, C., Perren, B., Roberts, S., de Boer, A. M., 2016: Sea ice led to poleward-shifted winds at the Last Glacial Maximum: the influence of state dependency on CMIP5 and PMIP3 models, Clim. Past, 12, 22412253, doi: /10.5194/cp-12-2241-2016)
*Thank you for pointing us to this interesting result. We included a brief discussion of the Southern Ocean westerly wind disagreement as follows: 'In the Southern Ocean the band of large disagreement in westerly wind speeds reflects the large uncertainty in the simulated position of the southern hemisphere jet stream, both in the pre-industrial and the LGM. This disagreement is reinforced by the fact that shifts in the jet position between pre-industrial and LGM also depend on the simulated expansion of sea-ice (Sime et al. 2016; Fig. A3).'*

Page 6, lines 24-26: "Regarding the dynamical inconsistencies, it is important to note that the CORE forcing itself is a mixture of reanalysis and 25 observational data products and as such not dynamically consistent." – This sentence is phrased in a way that makes me feel like the CORE

forcing has been previously discussed in the paragraph, which is not the case. Consider rephrasing or adding a sentence 'Our dataset has been adapted to be an extension of the CORE forcing.' (or similar) before this sentence.
*We added 'for which our dataset is optimized' in parentheses after the mention of CORE forcing.*

Page 8, lines 2-3: "The attribution of the model spread to specific processes is beyond the scope of this article [...]" – Yet the authors do attempt to provide these explanations for several variables. For a user of the dataset, I think it would be useful to know where the model spread originates from, and while reading the article, I found it somewhat frustrating that some of these explanations were missing (see my comment for page 5, lines 18-20). I feel that, in those cases where that information is available from other PMIP3 research, it is something that the authors could provide without additional analysis. For those variables that explanations for the model spread are given, there are generally no motivations for these explanations, for example in the form of a cited paper or a confirming supplementary figure (see e.g. page 4, lines 14-15; and page 5, lines 25-26, which could both be confirmed with a figure of model spread in sea-ice cover). Based on the fact that some explanations are indeed already given, I would advise to rephrase this sentence to 'The complete attribution of the model spread [...]' , whether or not any further explanations can be added.
*We added the word 'complete' as suggested as well as including the new model spread figure for ice cover, Fig. A3. We refer to this new figure in Sect 3.2, 3.5 and 3.6.*

Page 8, line 5: "Finally, there is no other way [...]" – I would suggest expressing this a bit more cautiously, by changing to Finally, there is currently no other way [...]
*Corrected as suggested by adding 'currently'.*

**Technical corrections**
Page 3, line 9: "yearly" – annual
*Corrected as suggested*

Page 3, line 19: "The SSS fields is [...]" – The SSS fields are [...]
*Corrected as suggested*

Page 3, line 35: "under the assumption of an unchanged spatial and temporal variability" – remove an
*Corrected as suggested*

Page 4, lines 6 and 8: "2-3 kg kg−1" and "1-2 kg kg−1" – According to the colourbar in Fig. 1, the magnitude of the specific humidity anomaly and spread is 10-3 kg kg−1 .
*Corrected as suggested*

Page 4, line 23: "symmetrically" – symmetrical, i.e. a symmetrical spatial pattern, though it is not entirely clear what symmetrically refers back to here. Consider either changing to symmetrical or rephrasing the sentence.
*Corrected to symmetrical*

Page 5, line 24: The temperature anomalies of 10K and ~2.5K are presumably supposed to be negative.
*Corrected as suggested by adding negative signs*

[revised manuscript text omitted]